# LEARNING A COMPACT, PARCEL-INDEPENDENT REPRESENTATION OF THE FMRI FUNCTIONAL CONNECTIVITY

## ABSTRACT

Functional connectivity in functional magnetic resonance imaging (fMRI) data is often calculated at the level of area parcels. Given the data's low-dimensional nature, we posit a substantial degree of redundancy in these representations. Moreover, establishing correspondence across different individuals poses a significant challenge in that framework. We hypothesize that learning a compact representation of the functional connectivity data without losing the essential structure of the original data is possible. Our analysis, based on various performance benchmarks, indicates that the pre-computed mapping to low-dimensional latent space learned from the functional connectivity of one dataset generalizes well to another with both linear and non-linear autoencoder-based methods. Notably, the latent space learned using a variational autoencoder represents the data more effectively than linear methods at lower dimensions (2 dimensions). However, at higher dimensions (32 dimensions), the differences between linear and nonlinear dimensionality reduction methods diminish, rendering the performance comparable to the parcel space representation with 333 dimensions. Our findings highlight the potential of employing an established transformation to obtain a low-dimensional latent representation in future functional connectivity research, thereby solving the correspondence problem across parcel definitions, promoting reproducibility, and supporting open science objectives.

## 1 INTRODUCTION

Spontaneous brain activity, as measured using the blood-oxygen-level dependent (BOLD) signals estimated with functional magnetic resonance imaging (fMRI), demonstrates a high correspondence between functionally homotopic brain regions (Biswal et al., 1995). Researchers have measured resting-state functional connectivity to delineate the functional organizations of the brain (Bijsterbosch et al., 2020; Eickhoff et al., 2015). Many studies omitted the spatial topography and described the brain organization as abstract nodes and edges, giving the pairwise "functional connectome" matrix and the associated network graph representations (Bassett & Sporns, 2017; Bullmore & Sporns, 2009; Cheirdaris, 2023; Rubinov & Sporns, 2010; Sporns et al., 2004). Many node definitions exist and vary in number from less than 100 to 1000 area parcels in an atlas (Arslan et al., 2018; Craddock et al., 2012; Gordon et al., 2016; Schaefer et al., 2018; Shen et al., 2013). In addition to the lack of consensus in node definition, recent work has demonstrated that the optimal definition of nodes can vary across individuals (Gordon et al., 2017b) and populations (Han et al., 2018; Myers et al., 2024). Secondly, recent studies showed that only a few dimensions can capture the majority of variance in functional connectivity data (Bolt et al., 2022; Gotts et al., 2020; Margulies et al., 2016; Snyder et al., 2022), potentially due to the high spatial correlation in the cerebral cortex (Pang et al., 2023; Shinn et al., 2023). Therefore, we explored the possibility of learning a pre-computed transformation that maps the functional connectivity seed maps (spatial patterns of connectivity from one node to the rest of the brain) to a low-dimensional latent representation with little compromise in preserving the key information for downstream analyses.

We obtained low-dimensional embeddings (2, 4, 32 dimensions) from the high-dimensional functional connectivity seed map (59412 dimensions) data using a few variants of autoencoders(Rumelhart et al., 1986) including the conventional autoencoder(Rumelhart et al., 1986), vari-

ational autoencoder(Kingma & Welling, 2022), $\beta$-variational autoencoder ($\beta$-VAE)(Higgins et al., 2017) , and adversarial autoencoder(Makhzani et al., 2016). We also compared the results with linear dimensionality reduction methods using Principal Component Analysis (PCA) and Independent Component Analysis (ICA). We explored the extent to which key features in the original data could be retained using these low-dimensional embeddings, and how it varied across dimensionality reduction methods and the number of dimensions. We adopted multiple performance metrics and used the parcel connectome with the Gordon parcellation (Gordon et al., 2016) of 333 nodes (a.k.a. 333 dimensions) as a reference benchmark.

## 2   RELATED WORK

Our current work is most similar to the representation of functional connectivity into principal gradients with diffusion embedding and other non-deep-learning-based methods (Langs et al., 2015; 2010; Margulies et al., 2016; Vos de Wael et al., 2020). However, learning the low-dimensional manifold in individuals and aligning them requires matched parcels across subjects using Procrustes analysis (Langs et al., 2015) and can be computationally expensive. Even more detrimentally, if the manifold spaces (e.g. across development or clinical conditions) differ substantially, the alignments may provide output that is not meaningful (Vos de Wael et al., 2020). Furthermore, this approach does not provide a backward projection to the original space from the embeddings to give an intuitive demonstration of a walk in the latent space. More discussion about challenges in establishing correspondence across subjects is provided in Appendix A.1.

There are also numerous other deep-learning approaches to capture fMRI activity and connectivity with advanced deep neural network architectures(Caro et al., 2023; Zhang et al., 2020; Zhao et al., 2020; Zuo et al., 2023; Ryali et al., 2024; Qiang et al., 2021). However, many dealt with data in an abstract node format and did not respect the spatial topography in functional connectivity in the anatomical space. Moreover, most prior work tested the model with performance on a specific task (e.g. disease classification(Qiang et al., 2021; Zhao et al., 2020)) or sex classification (Ryali et al., 2024) rather than a range of metrics as we suggested here. The majority of them use data samples as the complete functional connectome (one per subject) rather than the functional connectivity seed maps from each node, as we propose here. Moreover, to our knowledge, only limited prior research has investigated the use of an unsupervised method to embed fMRI data into a latent space (Caro et al., 2023; Kim et al., 2021) for general-purpose visualization and analysis, or demonstrated the effect of traversing the latent space. None of these studies have applied such methods to functional connectivity seed maps or used an extremely low-dimensional bottleneck (2, 4, and 32 dimensions) as we do here, and only one tried to generalize to a different dataset(Caro et al., 2023).

Here, we aim to find a pre-computed transformation that maps new functional connectivity seed map data onto a low-dimensional space for visualization and analysis. This approach is easy to apply and addresses the correspondence dilemma across node definitions. Furthermore, analyses such as clustering can benefit from the computational advantage of a low-dimensional embedding space (Langs et al., 2016; Zhakubayev & Hamerly, 2022) (more discussion on this in Appendix A.2).

## 3   METHODS

### 3.1   NEUROIMAGING DATA

We used resting-state fMRI data from the Washington University 120 (WU120) (Power et al., 2014) and the Human Connectome Project (HCP) (Glasser et al., 2016; Van Essen et al., 2012b) datasets for training and testing the models. Both datasets were collected from young adult subjects (19-35 years) while they were asked to fixate on a center cross on the screen in a 3-Tesla MRI scanner. Procedures were then applied to normalize intensity, correct for motion in the scanner, and transform the data onto a standard 32k-fsLR surface (Van Essen et al., 2012a). Details are available in the Appendix A.3. The WU120 dataset contains one resting-state session per subject for 120 subjects. Out of the 120 subjects from WU120, 100 subjects were selected as the training data, 10 as the validation data, and 10 as the test data. The HCP dataset contains 94 subjects from unrelated families, with two resting-state sessions for each subject. We used both sessions (Rest1 and Rest2) in the HCP dataset as the test data.

### 3.1.1 PREPARATION OF THE FUNCTIONAL CONNECTOME DATA

First, to make the functional connectome comparable across different nodal definitions, we used a functional connectivity seed map, where the values at each location in the cortical vertex represent its connectivity with a given seed region. This was calculated as the Pearson's correlation between the BOLD time series of the seed region and the BOLD time series at each vertex in the cerebral cortex surface (N = 59412 in the standard 32k-fsLR surface). This results in a matrix of $N_{nodes}$ seed maps, each with a dimension of 59412 x 1. To encourage diversity and variability in the training data while keeping computational demand manageable, we used the cortical vertices as seed regions ($N_{nodes}$ = 59412 per subject) and randomly sampled 10% of the seed maps for each subject (total training samples = 591200). In the test data, we used individual parcels from the Gordon parcellation (Gordon et al., 2016) as seed regions ($N_{nodes}$ = 333 per subject) for each subject.

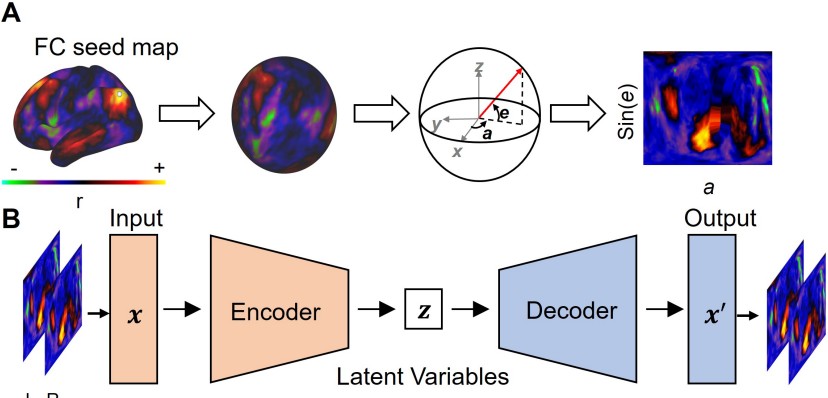

Figure 1: **Geometric reformatting and model architecture.** A) Geometric reformatting. The cortical distribution of fMRI activity is converted onto a spherical surface and then to an image by evenly resampling the spherical surface with respect to sin(e) and a, where e and a indicate elevation and azimuth, respectively. B) Architecture of VAE. Simplified block diagram of VAE model (upper panel in B). An encoder network samples latent variables given an input image under the inference model while a decoder network generates a genuine input image from under the generative model. Details of VAE model (bottom panel in B). The encoder and decoder networks both contain 5 convolutional layers and 1 fully-connected layer. Each input is a pair of 2-D images (left and right hemispheres).*Adapted with permission from Kim, J., Zhang, Y., Han, K., Wen, Z., Choi, M., & Liu, Z. (2021). Representation learning of resting state fMRI with variational autoencoder. NeuroImage, 241, 118423. Copyright 2021 by Elsevier Inc.*

For the autoencoder-based models, the input was passed on as a pair of 2-D images (left and right hemispheres) to take advantage of the convolutional layers of the model (Figure 1A, see details in Appendix A.4). This transformation would result in a small distortion such that the reconstructed 59412 x 1 seed maps from this 192 x 192 grid explain ˜99.9% variance from the original seed maps. To ensure fairness across algorithms, we passed the reconstructed seed maps (with distortion) instead of the original seed map from the 192 x 192 grid to PCA and ICA.

### 3.2 DIMENSIONALITY REDUCTION METHODS

### 3.2.1 AUTOENCODER-BASED METHODS

Autoencoders are neural networks designed to encode the input into a compressed representation, and then decode it back to a reconstructed input similar to the original one. The variational autoencoders (VAE) (Kingma & Welling, 2022) are especially useful in obtaining a smooth, continuous latent space for generating new data. The $\beta$-VAE variant (Higgins et al., 2017) is most effective in disentangling latent generative factors from images. Unlike t-SNE and diffusion maps, the AE-based models feature an intrinsic encoder for easy new data embedding and a decoder for data reconstruction. We adopted the same model architecture as described in (Kim et al., 2021; 2023; 2024) with five convolutional layers and one fully-connected layer in the encoder, and one fully-connected layer

and five convolutional layers in the decoder. The encoder transformed an fMRI seed map (a pair of left and right hemisphere images formatted to two 192 x 192 grids) into a probabilistic distribution of N latent variables (or a deterministic N latent variables in the case of a conventional autoencoder). Each convolutional layer conducted linear convolutions followed by rectifying the outputs as described by (Nair & Hinton, 2010). The first layer utilized $8 \times 8$ convolutions on inputs from each hemisphere and combined the results. Subsequent layers, from the second to the fifth, applied $4 \times 4$ convolutions to this combined output. Circular padding was employed at the azimuth boundaries, while zero padding was used at the elevation boundaries. A fully connected layer applied linear weighting to generate the mean and standard deviation for the distribution of each latent variable. The decoder replicated this structure in reverse, reconnecting the layers to recreate the fMRI seed map from a sample latent variable. Additional details on the model design and various autoencoder-based models can be found in Appendix A. 5 The VAE variant was optimized to reconstruct the input while constraining the distribution of every latent variable to be close to an independent and standard normal distribution. This is achieved by optimizing the encoding parameters, $\phi$, and the decoding parameters, $\theta$, to minimize the loss function below:

$$L(\phi, \theta \mid x) = \| x - x' \|_2 + \beta \cdot D_{KL} \left[ N(\mu_z, \sigma_z) \parallel N(0, I) \right] \tag{1}$$

where $x$ is the input data from both hemispheres, $x'$ is the reconstructed data, and $N(\mu_z, \sigma_z)$ is the posterior distribution, $N(0, I)$ is the prior distribution. $D_{KL}$ measures the Kullback-Leibler (KL) divergence between the posterior and prior distributions, and $\beta$ is a hyperparameter balancing the two terms in the loss function. A $\beta < 1$ places less emphasis on the KL divergence and focuses more on reconstruction, while a $\beta > 1$ places a higher emphasis on KL divergence, enforcing stricter regularization of the latent space. When $\beta = 0$, the loss function only depends on the reconstruction error and is thus the conventional autoencoder. Instead of using the KL-divergence loss for regularization, the adversarial autoencoder used a separate discriminator network (3-layers, with the first two with a leaky ReLU activation function with a negative slope angle of 0.2 and the last one with a sigmoid activation function) for regularization (Makhzani et al., 2016).

Models were trained with stochastic gradient descent with a batch size of 128, initial learning rate of $1 \times 10^{-4}$, and 50 epochs with random data selection in each batch. An Adam optimizer (Kingma & Ba, 2014) was implemented, and the learning rate decayed by a factor of 10 every 20 epochs for the AE and VAE models. Final hyperparameters including the number of latent dimensions and the beta value were determined by the trade-off between KL divergence and reconstruction loss on the validation data (See Appendix A.6). The model was trained in Python 3.8 using PyTorch (v2.1.2+cu118) using a server with an NVIDIA A100 GPU (40 GB memory). A brief demonstration of model performance variability using different subsets of training data is provided in Appendix A.7.

### 3.2.2 LINEAR DIMENSIONALITY REDUCTION METHODS

We chose principal component analysis (PCA) and independent component analysis (ICA) as alternative linear dimensionality reduction methods. PCA uses Singular Value Decomposition of the data, keeping only the most significant singular vectors to project the data to a lower dimensional space. ICA attempts to decompose the data into a set of independent spatial maps. For memory management, we performed PCA with incremental PCA (IPCA) from the Scikit-learn package (v1.3.2) in Python 3.8 (Golub & Loan, 2013; Ross et al., 2008). IPCA builds a low-rank approximation for the input data using a memory amount that is independent of the number of samples. ICA was performed on the reduced data using the first 100 PCs with FastICA in Scikit-learn.

### 3.3 ASSESSING THE QUALITY OF EMBEDDINGS

### 3.3.1 RECONSTRUCTION PERFORMANCE

Reconstruction performance was calculated with $\eta^2$, namely, the fraction of variance in the original seed map accounted for by variance in the reconstructed seed map on a point-by-point basis (Cohen et al., 2008). $\eta^2$ ranged from 0 (no similarity) to 1 (identical) and is formally defined by:

$$\eta^2 = 1 - \frac{SS_{\text{Within}}}{SS_{\text{Total}}} = 1 - \frac{\sum_{i=1}^{n} \left[ (a_i - m_i)^2 + (b_i - m_i)^2 \right]}{\sum_{i=1}^{n} \left[ (a_i - \bar{M})^2 + (b_i - \bar{M})^2 \right]} \tag{2}$$

where $a_i$ and $b_i$ represent the values at position $i$ in maps $a$ and $b$, respectively. $\mu_i$ is the mean value of the two images at position $i$, $(a_i + b_i)/2$. $\bar{M}$ is the grand mean value across the mean image $m$.

Self-connectivity at all positions was excluded. This similarity matrix is sensitive to the difference in $a$ and $b$ in scales and offsets. For convenience, we used the inversely formatted seed maps from the 2D image in Figure 1A as the reference ground truth data ($\eta^2 > 0.99$ to the original data) to be compared with the reconstructed data from latent representations. Note that perfect reconstruction is not necessarily desirable because the original data might be noisy, and the reconstructed data could potentially represent a "denoised" version of the data. We calculated two additional reference measures: the first one is the $\eta^2$ between the ground truth data in HCP Rest1 and HCP Rest2 of the same subject, which provides the noise ceiling of the reconstruction; the second one is the $\eta^2$ between each subject to the rest 93 subjects in each session (averaged across the Rest1 and Rest2 sessions), which provides the null baseline of group average.

### 3.3.2 SEPARATION BY CANONICAL FUNCTIONAL NETWORKS

One key feature of functional connectivity is that nodes within the same functional network tend to possess similar seed maps (Yeo et al., 2011). The 333 parcels in the Gordon parcellation were grouped into 12 functional networks and one additional group (named "None") of 47 parcels which covered the low-SNR regions and cannot be confidently grouped into any of the 12 functional networks and omitted from this analysis (Gordon et al., 2016). We evaluated the segregation of seed maps from different functional networks with the silhouette index (SI) (Rousseeuw, 1987; Yeo et al., 2011), calculated as:

$$SI(i) = \frac{b_i - a_i}{max(a_i, b_i)} \tag{3}$$

where $b_i$ is the mean Euclidean distance between the current parcel $i$ to the parcels in the best alternative network, and $a_i$ is the mean within-network Euclidean distance of the latent. A 95% confidence interval was calculated by bootstrapping the individuals 1000 times.

### 3.3.3 INTRASUBJECT AND INTERSUBJECT VARIABILITY

To capture individual differences in functional network assignments (Gordon et al., 2017a;b), we ran a k-means clustering algorithm on the latent embeddings with the cluster centroid initialized to be the center of each of the 12 Gordon networks (Gordon et al., 2016). 12 clusters were optimized by minimizing Euclidean distances between latent embeddings (dimensions = 2, 4, 32) within a cluster. Intersubject variability was calculated as the average normalized Hamming distance (a.k.a. proportion of mismatches, with 0 indicating perfect match) from each subject to other subjects; the intrasubject variability was calculated as the average normalized Hamming distance between the two resting sessions of the same subject. The latent representations that capture the most individual differences need to exhibit low intrasubject reliability while simultaneously maintaining a high intersubject variability. We calculated the variability signal-to-noise ratio (vSNR) to quantify the relative magnitude of the two sources of variability (Langs et al., 2016). Specifically:

$$vSNR = \frac{\text{Intersubject variability}}{\text{Intrasubject variability}} - 1 \tag{4}$$

## 4 RESULTS

### 4.1 SINGLE LATENT TRAVERSAL

To understand how the changes in the magnitude of each latent dimension affect the reconstructed seed map appearance, we plotted the single latent traversals when the number of dimensions = 2 (Figure 2). We varied the magnitude of one latent dimension while keeping the other latent dimension fixed at zero. We observed patterns reminiscent of sensorimotor networks and association networks (Margulies et al., 2016; Sydnor et al., 2021), as well as task-positive to task-negative networks (Buckner et al., 2008; Fox et al., 2005; Raichle, 2015). Notably, reconstructed maps resembling distinct somatomotor and visual networks were only observable in VAE ($\beta = 20$). We also observed a gradual transition from somatomotor hand to somatomotor mouth networks when varying a single dimension in VAE ($\beta = 20$). The conventional AE has similar profiles in its first and second latent dimensions, suggesting an ineffective disentangling of generative factors. In summary, a single latent dimension in VAE ($\beta = 20$) seems to disentangle the most subtle details in functional connectivity.

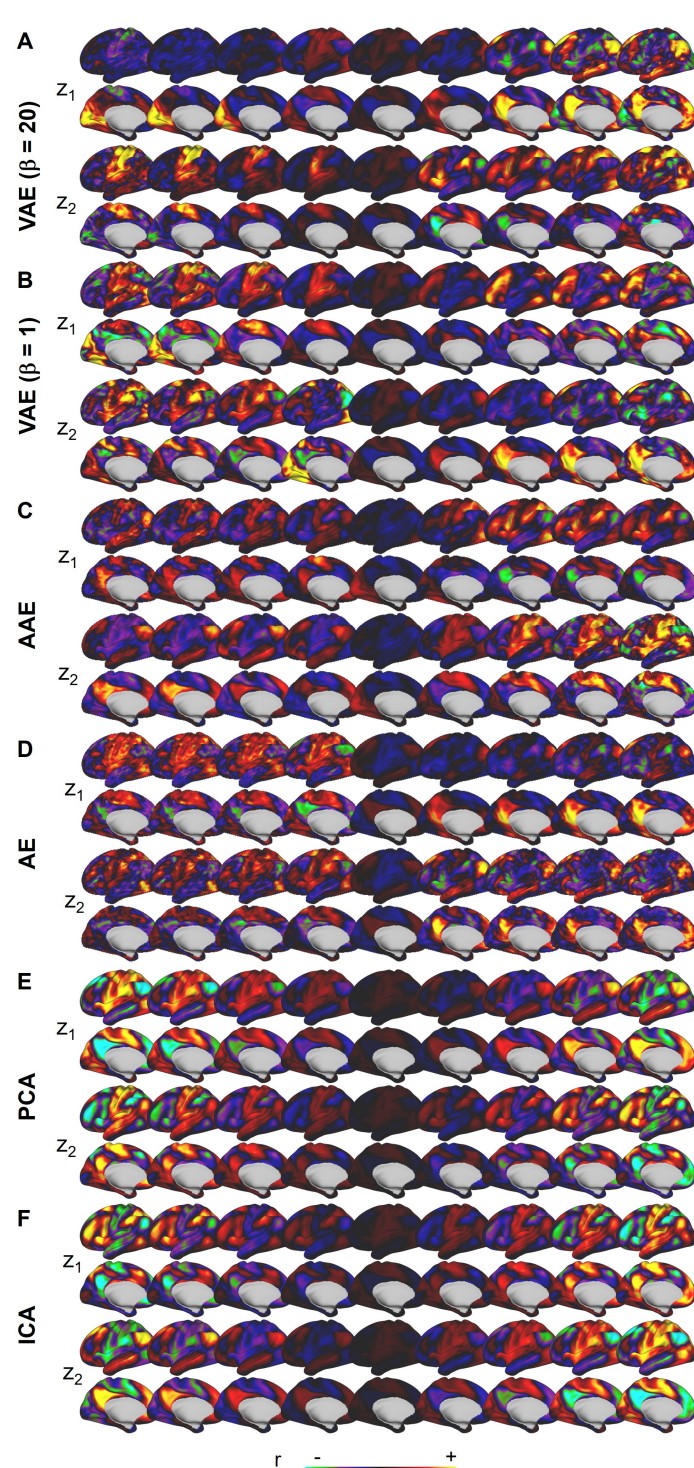

Figure 2: Single latent traversal. The reconstructed seed maps from latent values of one dimension varied in equal steps from one end to the other end and the other dimension was fixed to 0. A) VAE ($\beta = 20$), B) VAE ($\beta = 1$), C) AAE, D) AE, E) PCA, F) ICA.

## 4.2 RECONSTRUCTION PERFORMANCE

Next, we examined the reconstruction performance for parcel seed maps (functional connectivity from an area parcel defined from an atlas (Gordon et al., 2016)) to examine how effectively different dimensionality reduction methods retain information. 10 test subjects in the WU120 dataset and

both Rest1 and Rest2 sessions for the 94 subjects in the HCP dataset were chosen to test the out-of-sample and out-of-distribution generalization, respectively. Figure 3A shows reconstructed seed maps from one example parcel (parcel 15 in the medial visual cortex). Overall, the mean reconstruction performance across all parcels in each individual was similar across methods, with AE-based latent representations providing better reconstruction performance when the latent representation had only 2 dimensions (Figure 3B) and linear methods providing marginally better reconstruction performance at 32 dimensions (Figure 3D). In all cases, the reconstruction performance from the latent representations was on average higher than the null baseline, suggesting that the latent representations captured individual-specific features in additional to group-average features. With 32 dimensions, the reconstruction performance was approaching the noise ceiling (for the normalized reconstruction performance relative to the individuals' noise ceiling see Appendix A.9).

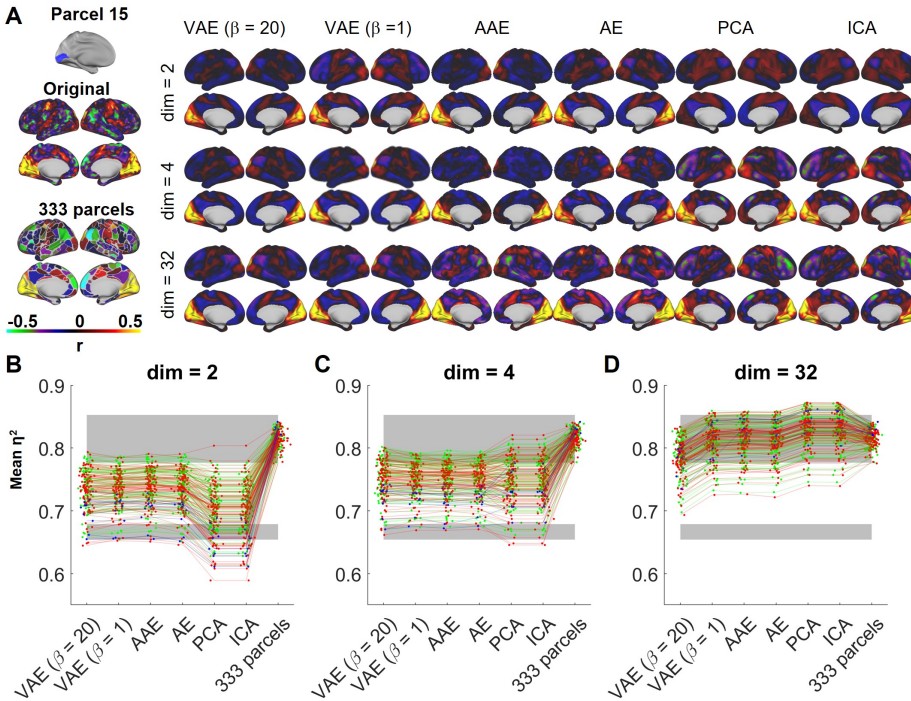

Figure 3: **Reconstruction performance in the test set.** A) Visualization of the original and reconstructed seed maps from parcel 15 in an example subject (subject 111). B-D) The reconstruction performance for all subjects in the HCP and WU120 test datasets. Reconstruction Green line: HCP Rest1, Red line: HCP Rest2, Blue line: WU120. Gray shaded area: mean and standard deviation of the noise ceiling and the null baseline. N.B.: 333 parcels always have 333 dimensions and were repeatedly displayed in all three panels as a reference. VAE = variational autoencoder. AAE = adversarial autoencoder. AE = autoencoder. PCA = principal component analysis. ICA = independent component analysis.

## 4.3 Separation by Canonical Functional Networks

Seed maps from area parcels belonging to the same functional network should be similar to each other. We asked whether the functional network clusters defined in adult group-average data (Gordon et al., 2016) were preserved in the low-dimensional embeddings (examples for dimension = 2 in Figure 4). We quantified the distinguishability from the best alternative network for each of the 333 parcels with the silhouette index (SI) for the average latent embeddings of Rest1 sessions seed maps across 94 HCP subjects. First, we found the separation by functional networks in the parcellated functional connectome (dimension = 333, SI = 0.158, see Appendix A.10). Even at dimension =2, functional network segregation in the latent space was evident across all dimensionality reduction methods, particularly in the separation of sensorimotor networks from association networks (with the exception of AE). However, the distinction within the sensorimotor networks (visual, somato-

motor, auditory) was far more pronounced in the VAE and AAE latent spaces, especially with VAE ($\beta = 20$). Another interesting observation was that the default mode network (colored red in Figure 4A) appeared to separate into two distinct clusters for the VAE ($\beta = 20$) latent space (See Appendix A.11). At two dimensions, the average SI for VAE ($\beta = 20$) was 0.086 (95% CI: [0.066, 0.099]), higher than the VAE ($\beta = 1$) (0.044, 95% CI: [0.015, 0.063]), AAE (0.035, 95% CI: [0.006,0.054]), AE (0.014,[-0.018,0.030]), PCA ($7 \times 10^{-4}$, 95% CI: [-0.016, 0.009]) and ICA (-0.010, 95% CI: [-0.028, 0.001]). When the latent dimensions were 4 or 32, the average SI can sometimes be higher than that in the parcel connectome (Table 2). Among the models tested, VAE ($\beta = 20$) with four dimensions (SI = 0.213, 95% CI: [0.197,0.221]) best separates the parcels according to the functional network labels (A.10). Notably, functional connectivity networks have hierarchical organization (Betzel & Bassett, 2017; Urchs et al., 2019), and the current functional network labels (Figure 4A) only represent one popular scale of investigation. The reduction in SI from dimension = 4 to dimension = 32 for some methods (A.10) might be due to the more pronounced separation of network sub-components at a relatively higher-dimension latent space.

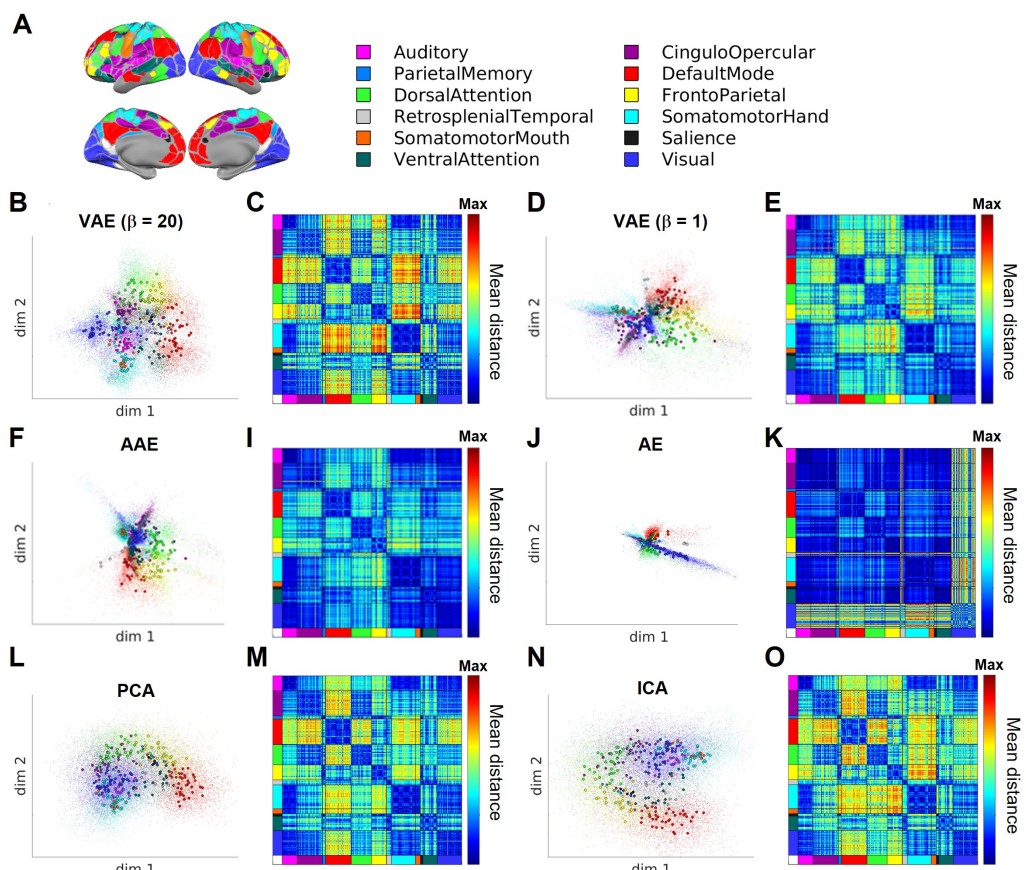

Figure 4: **Separation by canonical functional networks.** A) Gordon network assignments. B/D/F/J/L/N) The functional connectome 2-dimensional embeddings. Small dots represent data from individual subjects and large circles represent data averaged across 94 subjects. C/E/I/K/M/O) The mean Euclidean distance between the latent representations of the average across 94 subjects. VAE = variational autoencoder. AAE = adversarial autoencoder. AE = autoencoder. PCA = principal component analysis. ICA = independent component analysis.

## 4.4 INTRASUBJECT AND INTERSUBJECT VARIABILITY

Despite the largely consistent topography of functional networks (Damoiseaux et al., 2006; Gratton et al., 2018), individual differences in functional network assignment are evident in prior studies (Gordon et al., 2017b;a; Gratton et al., 2018). We tested how well the latent representations preserve

these individual differences. First, we examined the ability of the low-dimensional representation to capture the intersubject variability while preserving the similarity of intrasubject data in individual parcels. We used an example parcel (parcel 213) with a large variability. We found that the relative positions of the embeddings for different subjects were consistent across the two resting scan sessions (Rest1 and Rest2) in the VAE ($\beta = 20$) latent space (Figure 5). The two subjects at the extremes demonstrated very distinct seed map profiles (Figure 5). Next, we applied a k-means clus-

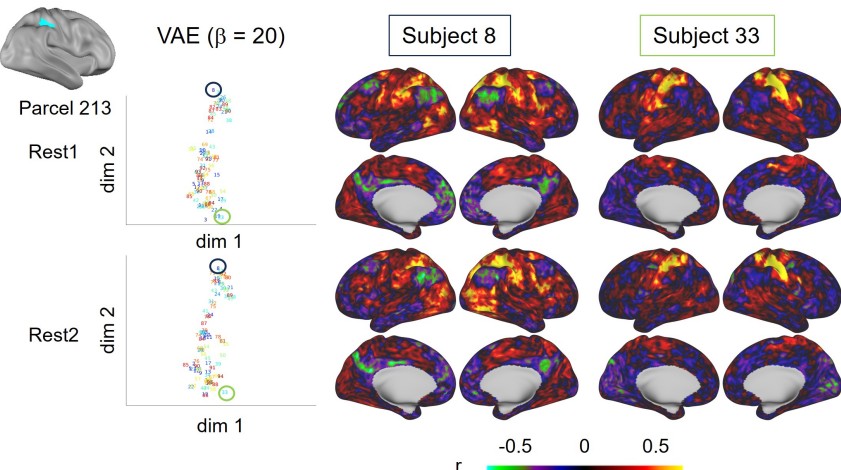

Figure 5: **Interindividual variability of an example parcel across sessions across 94 subjects in the VAE ($\beta = 20$) latent representation, at dimension = 2.** Each data point is a subject indicated by the numbers 1 to 94. The seed maps from parcel 213 for subject 8 and subject 33 in Rest1 and Rest2 were shown as an example.

tering with k = 12 to the low-dimensional embeddings for the 94 subjects in HCP across Rest1 and Rest2 sessions. We calculated the intrasubject and intersubject variability using Hamming distance (proportion of mismatch in the 333 parcel assignments) and vSNR as described in **section 3.3.3**. We found that independent of the dimensionality reduction method used, the intersubject variability was higher than the intrasubject variability by about 50%, suggestive of the preservation of individual specificity in the functional connectome in the latent spaces (See Table 3 in Appendix A.12). Across all subjects, the within-subject network assignment was more similar than the between-network subject assignment (Figure 6, also see Appendix A.12). vSNR was among the highest for VAE ($\beta = 20$) across 2, 4 and 32 dimensions. Furthermore, we tested whether different sessions of the same individual can be successfully identified from a list of subjects and found that the identification accuracy using the latent embeddings at various dimensionalities closely approximated the fingerprinting accuracy using the parcel connectome, especially when the scan length was long enough ($> 20$ min). One exception is the AE embeddings which had a low identification accuracy (Details in Appendix A.13). Similarly, the prediction performance of age and sex from the different embeddings was similar and increased with the number of dimensions (Appendix A.14).

## 5 DISCUSSION

This work presents the learning of mappings of functional connectivity seed maps from vertex space to a low-dimensional latent space using data from one young adult dataset. Among different performance benchmarks, including the reconstruction performance, separation by functional networks, and the retention of individual specificity, the linear dimensionality-reduction methods PCA and ICA achieved similar performance to the nonlinear AE-based dimensionality-reduction methods, especially at higher dimensions. This is consistent with prior research finding comparable functional connectivity gradients with PCA, Laplacian eigenmaps, and diffusion mapping (Vos de Wael et al., 2020), suggesting that the functional connectivity data likely has a predominantly linear structure. Other supervised machine learning approaches also found comparable performance with deep neural networks and kernel regression (He et al., 2018), suggesting that more complex models are not always better. However, with only 2 dimensions, the VAE ($\beta = 20$) latent embeddings best and

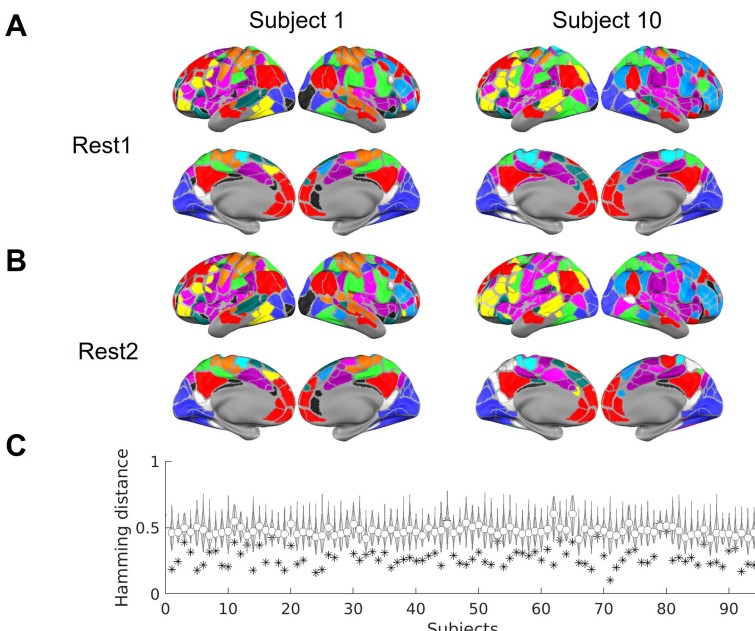

Figure 6: **Individual-specific network using k-means clustering with 12 networks (VAE beta = 20, dimension = 32).** The functional network assignment for Rest1 and Rest2 sessions based on k-means clustering on latent representations across of all parcels in all subjects and sessions for A) subject 1 and B) subject 10. C) The normalized Hamming distance for between-subject network assignments (violins) and within-subject network assignments (asterisks).

separates the fine nuisances across functional networks. It may make a useful visualization tool for simultaneously comparing the seed map profiles across sessions, individuals, and states independent of the definition of parcels. This could enable analyses such as community detection and behavioral phenotype prediction to be conducted in a much more manageable space, and allow efficient data sharing for big data initiatives (Horien et al., 2021).

We infer that our learned mappings will apply to new unseen datasets based on two lines of reasoning: 1) we demonstrated that the mappings were generalizable beyond the original training data, despite the differences in acquisition procedures and subjects; 2) extensive prior literature has observed low dimensionality in functional connectivity data (Gotts et al., 2020; Margulies et al., 2016; Snyder et al., 2022) with prominent patterns conserved across development (Dong et al., 2021; Xia et al., 2022). However, future studies could explicitly test this assumption in other populations, especially in developmental and clinical cohorts.

## 6    LIMITATIONS AND FUTURE WORK

We acknowledge that our training data is a small sample with a narrow profile of demographics. The training dataset could be extended to incorporate diverse demographics, acquisition parameters, developmental stages, etc. However, with the continuous, regularized latent space learned by a VAE, it is reasonable to assume that even if the exact seed map does not exist in the original dataset, as long as they are dominated by the same generative factors (as expected by the constraints of physical and biological properties), the latent space representation mapped using our existing model would still retain key information of the new dataset. Additionally, the VAE latent space lacks interpretability, unlike other methods that explore harmonic modes/eigenmodes in the brain based on geometric anatomy. Therefore, our model is a descriptive/phenomenological model rather than a mechanistic model. Moreover, our framework targets resting-state functional connectivity analyses and overlooks the temporal information in the neural data. Similar methods of representational learning of temporal information in the neural data exist in the literature(Caro et al., 2023; Kim et al., 2021), and we explained our motivation for embedding the functional connectivity data instead of time series data with more detailed comparison in appendix A.15.

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

# A    APPENDIX

## A.1    COMMUNITY DETECTION IN A SHARED LATENT SPACE HELPS ESTABLISH CORRESPONDENCE

One key pursuit of system neuroscience is to group the nodes in functional connectomes into functional networks or communities across states, individuals, and the lifespan (Betzel et al., 2014; Grayson & Fair, 2017; Mitra et al., 2017; Puxeddu et al., 2020; Tagliazucchi et al., 2013; Wig, 2017). A key hurdle for examining those communities is the establishment of correspondence across the different connectomes. Prior work has attempted this by detecting communities in individual connectomes and then matching the degree of overlap in their topography either through visualization (Gordon et al., 2017b) or using a Hungarian matching algorithm to minimize the Hamming distance (Langs et al., 2016). Alternatively, a multilayer network can be constructed by linking multiple functional connectomes as layers (Bassett et al., 2011; Betzel et al., 2019; Puxeddu et al., 2020), where community detection methods were then applied. Neither approach could be applied to functional connectomes with different nodal definitions optimized for individuals (Gordon et al., 2017b; Laumann et al., 2015) or populations (Han et al., 2018; Myers et al., 2024).

## A.2    THE NEED FOR IMPROVING COMPUTATIONAL EFFICIENCY

The idea of improving computational efficiency on clustering by first performing dimensionality reduction rather than using the original high-dimension data has proven effective in other domains(Zhakubayev & Hamerly, 2022). While the number of nodes (100-1000) and edges (4950-499500) might not seem big for an individual functional connectivity matrix, the total data space can become large when considering all individuals (e.g., for UK Biobank it would be 40000+ (Horien et al., 2021)), longitudinal sessions, different data modality, and time windows within a session (de Domenico, 2017; Muldoon & Bassett, 2016; Betzel et al., 2019). In addition, recently people combine across multiple datasets for life span studies (Sun et al., 2023) which can increase the sample size further.

The theoretical time complexity of the Louvain algorithm in single-layer modularity maximization is $O(n(log(n)))$ where $n$ is the number of nodes. We can approximately calculate the time complexity of the Louvain algorithm in multi-layer modularity maximization to be $O(N \times L(log(N \times L))$ where $N$ is the number of nodes per layer and $L$ is the number of layers. Here, we monitored the computational time for community detection with a multilayer network using the MATLAB code provided by (Betzel et al., 2019) with different combinations of $N$ and $L$. For each repetition, we randomly drew hyperparameters $\omega$ and $\gamma$ which control the strength of interlayer coupling and resolution scale, respectively. The experiments were conducted on a server equipped with two AMD EPYC 7713 64-core Processors, providing 128 cores and 256 threads, with a base clock speed of 2.56 GHz. Our empirical results matched the theoretical expectations (Table 1).

| N ( number of nodes) | L (number of layers) | Time in seconds to run 100 repetitions |
|---|---|---|
| 200 | 5 | 8.3 |
| 200 | 50 | 120 |
| 200 | 500 | 3265 |
| 333 | 5 | 17 |
| 333 | 50 | 380 |
| 333 | 500 | 6694 |

Table 1: Computational time monitoring for multilayer modularity maximization

On the other hand, other popular clustering algorithms like Gaussian Mixture models have a quadratic dependence on the number of dimensions due to the complexity of manipulating covari-

ance matrices. Therefore, even though there is no strong need to go from the parcel space (hundreds) to a low-dimensional latent space, it should still provide noticeable improvements.

### A.3 NEUROIMAGING DATA ACQUISITION AND PROCESSING DETAILS

#### A.3.1 WASHINGTON UNIVERSITY 120 (WU120) DATA

In summary, data was obtained from 120 healthy young adults (60 females, average age = 25 years, age range = 19–32 years). Participants were right-handed, native English speakers recruited from the Washington University community. Screening via self-report questionnaire ensured no history of neurological or psychiatric diagnosis, nor head injuries resulting in more than 5 minutes of unconsciousness. All participants provided informed consent, and the study was approved by the Washington University School of Medicine Human Studies Committee and Institutional Review Board. Data is available at https://openneuro.org/datasets/ds000243/versions/00001/file-display/00001

Structural and functional MRI data were obtained with a Siemens MAGNETOM Trio Tim 3.0-T Scanner and a Siemens 12-channel Head Matrix Coil. Structural imaging included a T1-weighted sagittal magnetization-prepared rapid acquisition gradient-echo (MP-RAGE) structural image was obtained [time echo (TE) = 3.08 ms, time repetition, TR (partition) = 2.4 s, time to inversion (TI) = 1000 ms, flip angle = 8°, 176 slices with $1 \times 1 \times 1$ mm voxels]. Functional scan slices were aligned parallel to the anterior commissure–posterior commissure plane of the MP-RAGE and centered on the brain using an auto-align pulse sequence protocol available in Siemens software. This alignment corresponds to the Talairach atlas(Talairach, 1988).

For the functional MRI data acquisition, subjects were instructed to relax while maintaining fixation on a black crosshair against a white background. Functional imaging used a BOLD contrast-sensitive gradient-echo echo-planar imaging (EPI) sequence (TE = 27 ms, flip angle = 90°, in-plane resolution = $4 \times 4$ mm). Full-brain EPI volumes (MR frames) of 32 contiguous, 4-mm-thick axial slices were obtained every 2.5 seconds. Additionally, a T2-weighted turbo spin-echo structural image (TE = 84 ms, TR = 6.8 s, 32 slices with $1 \times 1 \times 4$ mm voxels) in the same anatomical planes as the BOLD images was also captured to augment atlas alignment. The fMRI acquisition used Anterior→Posterior (AP) phase encoding. The number of volumes collected per subject ranged from 184 to 724, with an average of 336 frames (14.0 min).

Functional images were first processed to reduce artifacts including (1) correction of odd versus even slice intensity differences due to interleaved acquisition without gaps, (2) head movement correction within and across runs, and (3) across-run intensity normalization to a whole-brain mode value of 1000. Each individual's functional data was transformed into an atlas space using the MP-RAGE scan and resampled to an isotropic 3-mm atlas space(Talairach, 1988), using a single cubic spline interpolation(Lancaster et al., 1995).

Additional preprocessing mitigated high-motion frame effects in two iterations. The first iteration included: (1) demeaning and detrending, (2) multiple regression including whole-brain, ventricular cerebrospinal fluid (CSF), and white matter signals, and motion regressors derived by Volterra expansion and (3) a band-pass filter ($0.009Hz < f < 0.08Hz$). Temporal masks were created in this iteration to flag motion-contaminated frames, identified using framewise displacement (FD), calculated as the squared sum of the motion vectors (Power et al., 2012). Volumes exceeding FD $> 0.2$ mm and segments of fewer than 5 contiguous volumes were flagged for removal.

The data were then reprocessed in a second iteration, incorporating the temporal masks described above. This reprocessing was identical to the initial processing stream but ignored censored data. Data were interpolated across censored frames using least squares spectral estimation (Power et al., 2014) of the values at censored frames so that continuous data could be passed through the band-pass filter ($0.009Hz < f < 0.08Hz$) without contaminating frames near high motion frames. Censored frames were ultimately ignored during functional connectivity matrix generation.

Individual surfaces were generated from the structural images and functional data were sampled to surface space (Glasser et al., 2013). Following volumetric registration, left and right hemisphere anatomical surfaces were created from each subject's MP-RAGE image using FreeSurfer's recon-all processing pipeline (v5.0)(Fischl, 2012). This involved brain extraction, segmentation, white matter and pial surface generation, surface inflation to a sphere, and spherical registration of the subject's "native" surface to the fsaverage surface. The fsaverage-registered surfaces were then

aligned and resampled to a resolution of 164000 vertices using Caret tools (Van Essen et al., 2001) and down-sampled to a 32492 vertex surface (32k-fsLR). Functional BOLD volumes were sampled to each subject's individual "native" midthickness surface (generated as the average of the white and pial surfaces) using the ribbon-constrained sampling procedure available in Connectome Workbench (v0.84) and then deformed and resampled from the individual's "native" surface to the 32k-fsLR surface. The final time series were smoothed along the 32k-fsLR surface using a Gaussian smoothing kernel ($\sigma = 2.55$ mm).

### A.3.2 HUMAN CONNECTOME PROJECT (HCP) DATA

Data for the Human Connectome Project (Young Adult) was collected at Washington University in St. Louis and the University of Minnesota. The participants, healthy adults aged between 22 to 35 years, underwent high-resolution T1-weighted (MP-RAGE, TR = 2.4s, voxel size = 0.7×0.7×0.7mm) and BOLD contrast-sensitive imaging (gradient echo EPI, multiband factor 8, TR = 0.72s, voxels = 2×2×2mm) using a custom Siemens SKYRA 3.0T MRI scanner equipped with a custom 32-channel Head Matrix Coil. Sequences with both left-to-right (LR) and right-to-left (RL) phase encoding were employed, with each participant completing a single run in each direction over two consecutive days, resulting in four runs in total, two for Rest 1 and another two for Rest 2. Data is available at https://www.humanconnectome.org/study/hcp-young-adult/document/1200-subjects-data-release

Previous research has indicated that minimally pre-processed data does not sufficiently control for confounds such as subject head motion (Burgess et al., 2016). In addition, reliable associations between functional connectivity and behavior require sufficient low-motion functional connectivity data for each subject Gordon et al. (2017b); Laumann et al. (2015). A similar preprocessing method as before was implemented (Seitzman et al., 2020). First, to account for magnetization equilibrium and any responses evoked by the scan start (Laumann et al., 2015), the first 29.52 seconds or – 41 frames – of each resting-state run were discarded. Then, the functional data were aligned to the first frame of the first run using rigid body transforms, motion corrected (3D-cross realigned), and whole-brain mode 1000 normalized (Miezin et al., 2000). The data, with 2x2x2mm voxels, was then registered to the T1-weighted image and a WashU MNI atlas using affine and FSL transformations (Smith et al., 2004).

Further preprocessing of the resting-state BOLD data was applied to remove artifacts (Ciric et al., 2017; Power et al., 2014). This involved calculating frame-wise displacement (FD) to quantify motion between consecutive frames in fMRI data (Power et al., 2012) and artifact removal using a low-pass filter at 0.1 Hz to address respiration artifacts affecting the FD estimates (Fair, 2020; Siegel et al., 2017), along with a threshold for removing frames with FD greater than 0.04 mm after the low-pass filter (N.B. for section A.6, no frame removal was applied because this leaves different numbers of valid frames for each subject). For functional connectivity (FC) analysis preparation, regression of nuisance variables was performed, including 36 regressors: (1) 3 time series (whole-brain mean, mean ventricular CSF, mean white matter) with temporal derivatives from the Volterra expansion (12 total parameters)(Friston et al., 1996), and (2) 6 head motion parameters with temporal derivatives from the Volterra expansion (24 total parameters). Spatial masks of the gray matter, white matter, and ventricles were created from the T1-weighted images for each of the individual-specific regressors using Freesurfer 5.3 automatic segmentation (Fischl et al., 2002). Data segments shorter than 5 contiguous frames were excluded, and least squares spectral estimation was used for interpolation over the censored frames (Hocke & Kämpfer, 2009; Power et al., 2014). The final data were then bandpass filtered from 0.009 to 0.08 Hz, and censored frames were excluded from the time series (Seitzman et al., 2020). It is crucial to perform censoring and interpolation before filtering to prevent high-motion noise artifacts from smearing into adjacent frames.

Following that, the preprocessed BOLD time series data underwent surface processing, which involved using the ribbon-constrained sampling procedure in Connectome Workbench to sample the BOLD volumes to each subject's native surface and exclude voxels with a time series coefficient with a variation of 0.5 SDs above that of the mean of nearby voxels (Glasser et al., 2013). After being sampled to the surface, time courses were deformed, resampled, and smoothed with a Gaussian smoothing kernel (FWHM = 4mm, $\sigma = 1.7$). Connectome Workbench was then used to combine these surfaces with volumetric subcortical and cerebellar data into the CIFTI format, generating full-brain time courses while excluding non-gray matter tissue (Glasser et al., 2013).

### A.4 GEOMETRIC REFORMATTING INTO 2-D IMAGES

The geometric reformatting procedure was done in four steps. First, the seed maps were mapped to the cortical surface using their coordinates in the 32k-fsLR mesh of the left and right hemispheres (32492 vertices per hemisphere with some of them empty due to the presence of the medial wall). Then, the surfaces in each hemisphere were inflated to a sphere using FreeSurfer (Fischl, 2012). After that, we used cart2sph.m in MATLAB to convert its Cartesian coordinates (x,y,z) to spherical coordinates (a,e), which reported the azimuth and elevation angles in a range from $-\pi$ to $+\pi$ and from $-\pi/2$ to $+\pi/2$, respectively. Lastly, we defined a $192 \times 192$ grid to resample the spherical surface with respect to azimuth and sin(elevation) such that the resampled locations were uniformly distributed at approximation.

### A.5 AUTOENCODER NETWORK ARCHITECTURE

In the encoder network, the size of the output image of each convolutional layer (from left to right) is 96x96x64 (32 channels per hemisphere), 48x48x128, 24x24x128, 12x12x256, and 6x6x256; for the decoder network, 6x6x256, 12x12x256, 24x24x128, 48x48x128, and 96x96x64 (32 channels per image), from left to right. The convolution operations are defined as 1: convolution (kernel size=8, stride=2, padding=3) with rectified nonlinearity, 2-5: convolution (kernel size=4, stride=2, padding=1) with rectified nonlinearity, 6: fully-connected layer with re-parametrization, 7: fully-connected layer with rectified nonlinearity, 8-11: transposed convolution (kernel size=4, stride=2, padding=1) with rectified nonlinearity, 12: transposed convolution (kernel size=8, stride=2, padding=3). The code implementation was adapted from https://github.com/libilab/rsfMRI-VAE/.

The adversarial network had a separate discriminator component and the adversarial loss was combined with reconstruction loss for regularization of the latent space(Makhzani et al., 2016). The code implementation was adapted from https://github.com/eriklindernoren/PyTorch-GAN/blob/master/implementations/aae/aae.py.

### A.6 DETERMINATION OF HYPERPARAMETERS

#### A.6.1 NUMBER OF LATENT DIMENSIONS

While it is desirable to have the lowest dimension possible for best computational efficiency, a dimension that is too small might not capture all crucial structures in the original data. Based on prior literature, we believed that a dimension greater than 40 would provide little additional information (Gotts et al., 2020; Margulies et al., 2016). We validated this in our data by evaluating the reconstruction loss and KL divergence in models with a range of dimensions (2, 4, 8, 16, 32, 64, 96, 128, 256) at $\beta = 1$ on the validation data (Figure 8A). We found that 32 dimensions were close to the elbow of the tradeoff between reconstruction performance and KL divergence, so we only tested models with latent dimensions 2, 4, and 32, with the reasoning that a latent dimensionality beyond 32 is likely to provide little improvement in reducing the reconstruction loss but would keep increasing the KL divergence. The same metric for dimensions = 2,4,8,16,32 in $\beta = 20$ was plotted as a comparison. We applied the same latent dimensions to the PCA and ICA approaches.

#### A.6.2 VAE BETA

As above, we explored a range of beta values from X to Y and found that $\beta = 20$ provided a good balance between reconstruction performance and KL divergence on the validation data (Figure 8B).

### A.7 MODEL VARIABILITY

One limitation is that the performance of the models could be biased by the samples selected to train the model. Here, we demonstrate the model variability with an example VAE at $\beta = 20$ with 5-fold cross-validation in the training data (Figure 9). The 100 subjects were divided into 20-subject groups. For each fold, we picked 4 groups to train the model and test on the same 10 subjects outside the 100 subjects as described in section A.3.1. We can see that the models achieved very similar performance on the training data, but had a larger variability in how well it generalize to an unseen validation set.

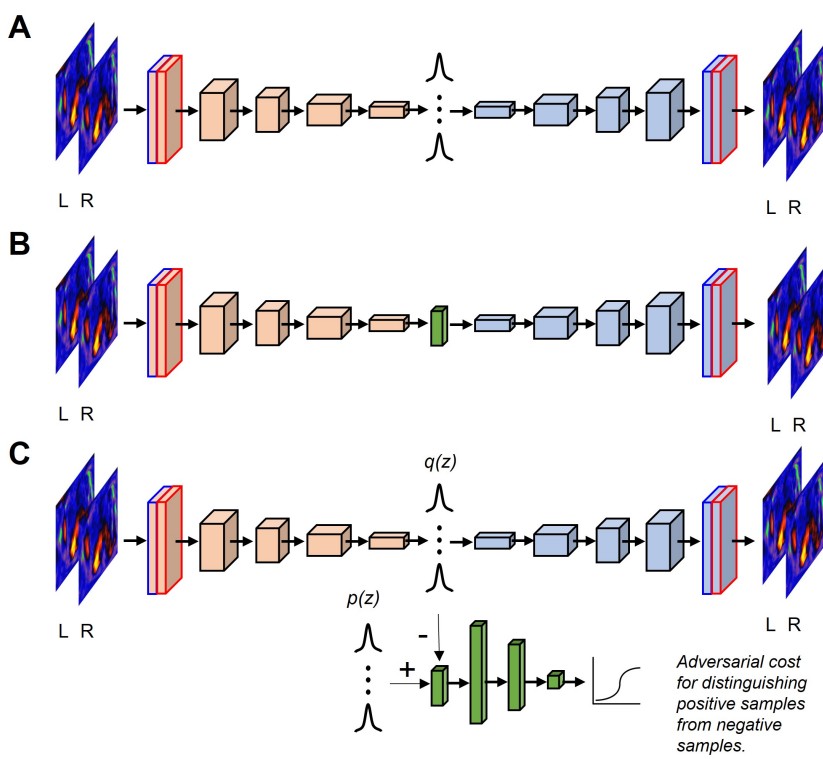

Figure 7: Autoencoder variants. A) Variational autoencoder. B) Autoencoder. C) Adversarial autoencoder. Blue and red boxes stand for the input images from left and right hemispheres, respectively. *A-C: Adapted with permission from Kim, J., Zhang, Y., Han, K., Wen, Z., Choi, M., & Liu, Z. (2021). Representation learning of resting state fMRI with variational autoencoder. NeuroImage, 241, 118423. Copyright 2021 by Elsevier Inc. C: Adapted from Makhzani, A., Shlens, J., Jaitly, N., Goodfellow, I., Frey, B., ICLR 2016. Adversarial Autoencoders.*

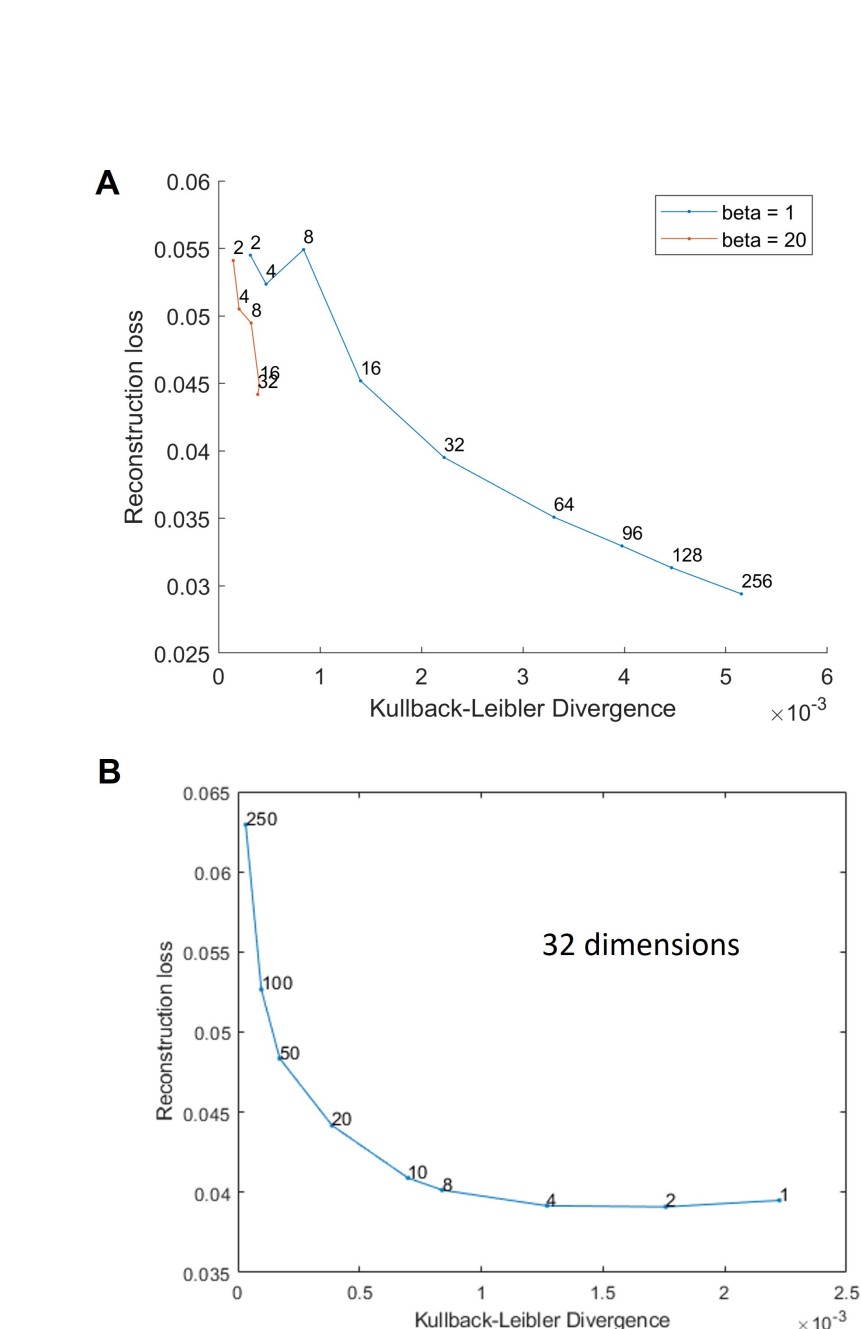

Figure 8: **Hyperparameter tuning.** A) Reconstruction loss and Kullback-Leibler divergence for different numbers of latent dimensions in the validation data. B) Reconstruction loss and Kullback-Leibler divergence for different $\beta$ values in the validation data.

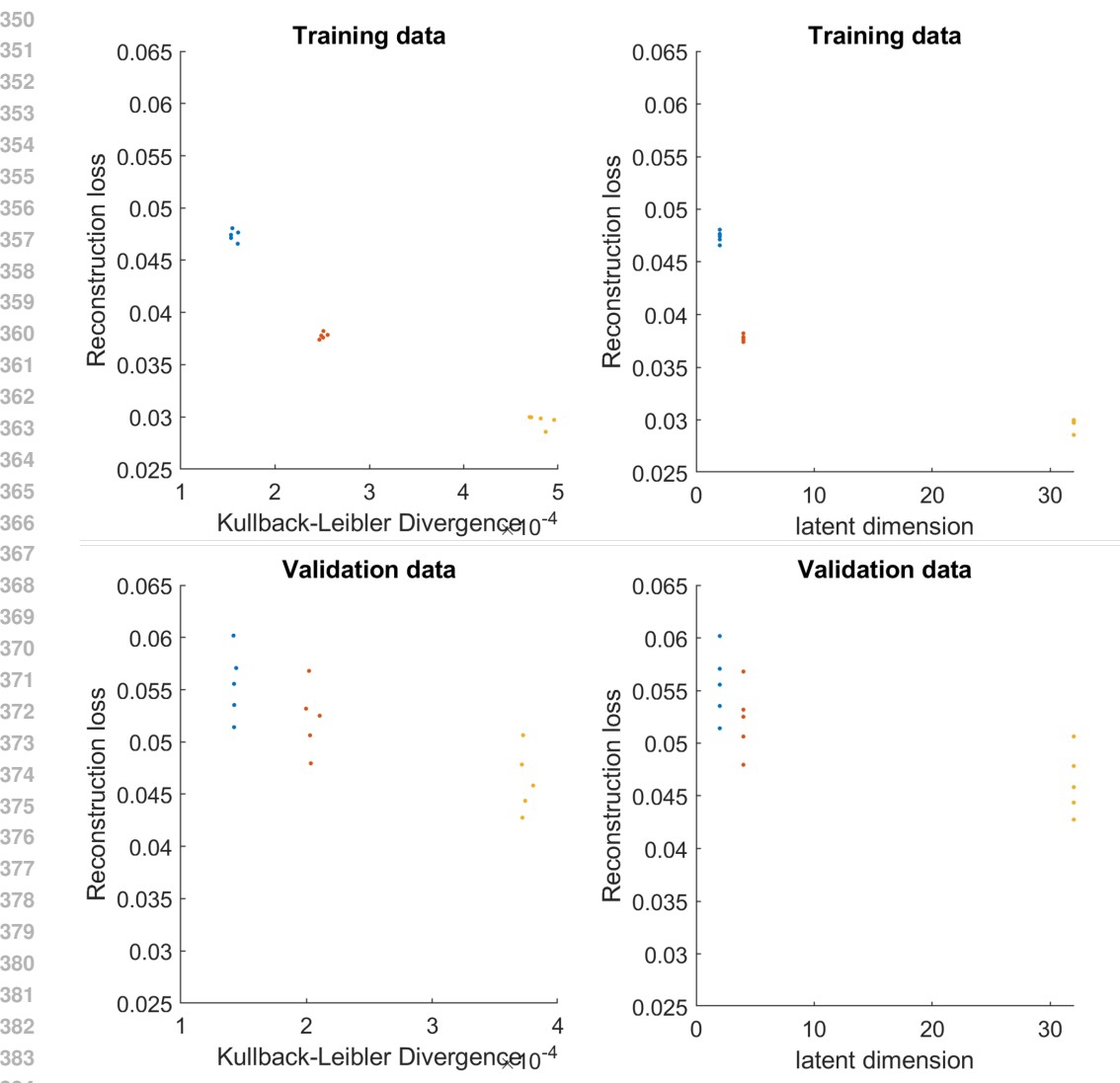

Figure 9: **5-fold cross-validation.** A)Reconstruction loss against KL-divergence for training data. B)Reconstruction loss against KL-divergence for test data. C-D) Same as A-B for validation data.

### A.8 PSEUDO-CODE FOR ANALYSIS PIPELINE USING THE CONNECTOME LATENT REPRESENTATIONS

Below is the pseudo-code for generating the seed maps from each parcel and then embed them in a low-dimensional latent space.

```
FOR each data session IN all_subjects:
    FOR each parcel i:
        // Compute the seed map for the parcel
        ParcelSeedMap = correlation(ParcelTimeSeries(:, i),
VertexTimeSeries(:, i))
        // Encode the seed map to obtain latent representation
        ParcelSeedMapLatents = Encoder(ParcelSeedMap)
        // Concatenate the latent representations across all parcels
        Append ParcelSeedMapLatents TO ParcelSeedMapAllLatents
    END FOR
    // Save the concatenated latent representations for
clustering/prediction
    SAVE ParcelSeedMapAllLatents
END FOR
```

### A.9 RECONSTRUCTION PERFORMANCE NORMALIZED TO THE NOISE CEILING

Since there is some variability in noise ceilings (a.k.a. the $\eta^2$ between the two resting scan sessions for the same subject), we additionally plot the normalized reconstruction performance calculated by dividing the actual reconstruction performance ($\eta^2$) by the noise ceiling for each subject in the HCP dataset.

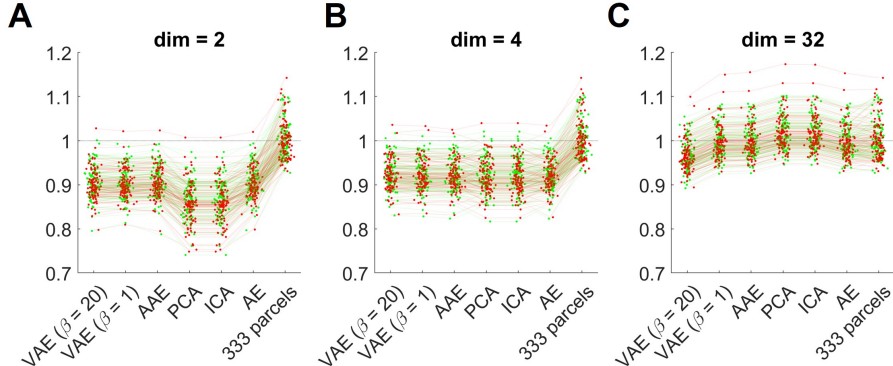

Figure 10: **Reconstruction performance in the HCP data.** A-C) The reconstruction performance for all subjects. Reconstruction Green line: HCP Rest1, Red line: HCP Rest2. Gray shaded area: mean and standard deviation of the noise ceiling and the null baseline. N.B.: 333 parcels always have 333 dimensions and were repeatedly displayed in all three panels as a reference. VAE = variational autoencoder. AAE = adversarial autoencoder. PCA = principal component analysis. ICA = independent component analysis. AE = autoencoder.

### A.10 SEPARATION BY CANONICAL FUNCTIONAL NETWORKS

We found that the average SI of the 333 parcel connectome is 0.158 (95% CI: [0.145, 0.163], Figure 11). Also, the average SI was higher in dim = 4 and dim = 32 than in dim = 2 (Table 2).

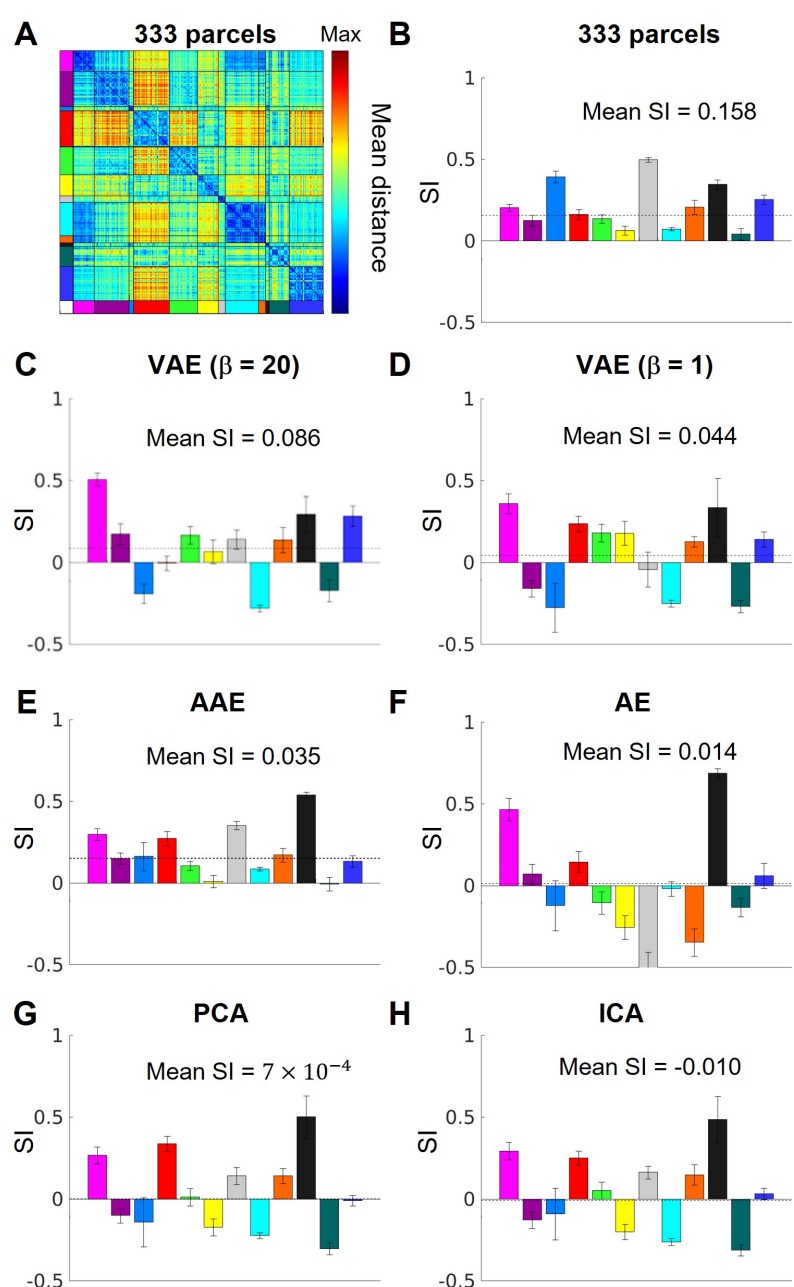

Figure 11: **Separation of networks A) Mean distance between area parcels in the latent embeddings with the 333 parcels (dimension = 333). B) Mean $\pm$ standard error SI for each functional network with the 333 parcels. C-H) Same as B but for the dimension = 2 across dimensionality reduction methods as shown in Figure 4.**

| Mean SI | VAE ($\beta$ = 20) | VAE ($\beta$ = 1) | AAE | AAE | PCA | ICA |
|---|---|---|---|---|---|---|
| dim = 2 | 0.086 [0.066, 0.099] | 0.044 [0.015, 0.063] | 0.035 [0.006, 0.054] | 0.014 [-0.018, 0.030] | $7 \times 10^{-4}$ [-0.016, 0.009] | -0.010 [ -0.028, 0.001] |
| dim = 4 | **0.213** [0.197, 0.221] | 0.193 [0.176, 0.200] | 0.181 [0.161, 0.188] | 0.177 [0.158, 0.184] | 0.173 [0.155, 0.181] | 0.173 [0.156, 0.182] |
| dim = 32 | 0.171 [0.158, 0.175] | 0.191 [0.177, 0.194] | 0.152 [0.136, 0.152] | 0.117 [0.094, 0.120] | 0.178 [0.164, 0.183] | 0.114 [0.104, 0.118] |

Table 2: **Silhouette index for different dimensionality reduction methods for the Gordon network clusters.**"dim" stands for dimensions.

### A.11 PROPERTIES OF DEFAULT SUB-CLUSTERS IN VAE LATENT SPACE (DIM = 2)

We applied a k-means clustering algorithm on the connectome latent representations from all default network parcels from the Rest1 session in all HCP 94 subjects in the 2-D VAE latent space (Figure 12A-B). The resultant cluster contains data from similar parcels across the subjects (Figure 12C) with some variability across subjects (Figure 10D). Cluster 1 seems to contain mostly parcels in the medial prefrontal cortex, the temporal cortex and the dorsolateral prefrontal cortex, while cluster 2 seems to contain mostly parcels from the inferior parietal cortex and the posterior cingulate cortex (Figure 12E). Subtle differences can be seen in the reconstructed seed maps from the centroids of the two clusters (Figure 12F). While the distribution of cluster 1 resembles the ventromedial and pregenual components of the default network, and the distribution of cluster 2 resembles the dorsolateral and parietal components of the default network as identified in an earlier study (Gordon et al., 2020), the biological and functional relevance of the two clusters identified here requires further investigation. However, the key takeaway of this analysis is that this eluded the possibility of identifying different subcomponents or individual/developmental-specific variants of existing functional connectivity networks using clustering of parcel connectome latent representations across subjects.

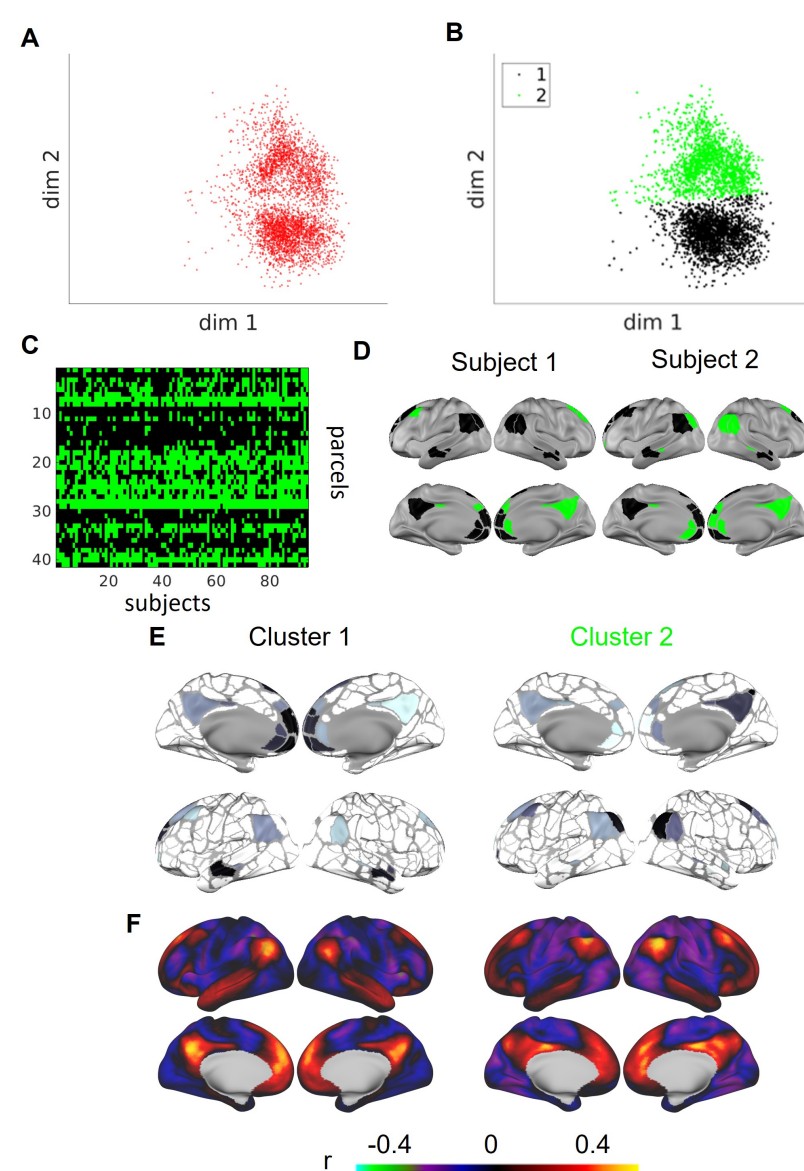

Figure 12: **Separation of DMN into subclusters.** A) DMN parcels across all subjects. B) Clustering the data in A into two sub-clusters with the k-means algorithm. C) The sub-cluster membership across parcels and subjects. D) The sub-cluster membership in two example subjects. E) The relative frequency of cluster membership across subjects (darker = more subjects) for the two sub-clusters. F) The reconstructed seed maps from the centroids of the two sub-clusters.

## A.12 K-MEANS CLUSTERING OF ALL PARCEL SEED MAPS IN THE LATENT SPACE

We illustrated the initial and final results of the optimization of cluster separation of the 12 functional networks using k-means clustering in Figure 13. Here we only show the Rest1 sessions of the 94 HCP subjects for visualization purposes, but the k-means algorithm was run on data from both Rest1 and Rest2 sessions. The k-means algorithm minimizes the squared distance between all the points to their closest cluster center by changing the cluster membership of the data points. More details can be seen in Table 3 and Figure 14-21.

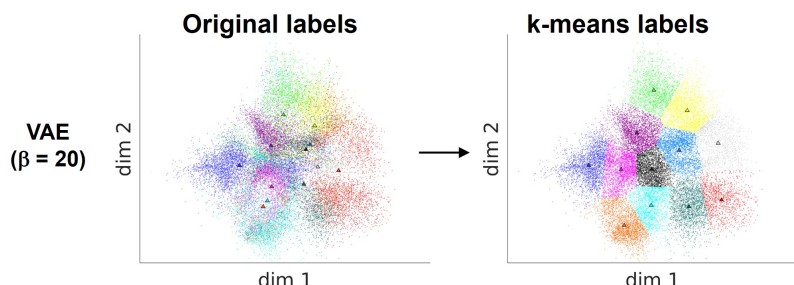

Figure 13: **k-means clustering in the latent space (2 dimensions).** (Left) Data from Rest1 in HCP 94 subjects with the original labels (dots) and centroid (triangle). (Right) Data from Rest1 in HCP 94 subjects with the optimized label after k-means clustering with 12 clusters.

| Intrasubject variability | dim = 2 | dim = 4 | dim = 32 |
|---|---|---|---|
| VAE ($\beta = 20$) | $0.39 \pm 0.07$ | $0.31 \pm 0.07$ | $0.29 \pm 0.08$ |
| VAE ($\beta = 1$) | $0.41 \pm 0.08$ | $0.40 \pm 0.08$ | $0.35 \pm 0.08$ |
| AAE | $0.40 \pm 0.08$ | $0.38 \pm 0.08$ | $0.45 \pm 0.10$ |
| AE | $0.39 \pm 0.08$ | $0.40 \pm 0.09$ | $0.54 \pm 0.10$ |
| PCA | $0.47 \pm 0.11$ | $0.38 \pm 0.10$ | $0.34 \pm 0.09$ |
| ICA | $0.48 \pm 0.10$ | $0.36 \pm 0.09$ | $0.27 \pm 0.05$ |
| 333 parcels | | $0.33 \pm 0.08$ | |
| Intersubject variability | dim = 2 | dim = 4 | dim = 32 |
| VAE ($\beta = 20$) | $0.60 \pm 0.03$ | $0.51 \pm 0.03$ | $0.48 \pm 0.03$ |
| VAE ($\beta = 1$) | $0.62 \pm 0.03$ | $0.60 \pm 0.03$ | $0.58 \pm 0.03$ |
| AAE | $0.60 \pm 0.03$ | $0.56 \pm 0.04$ | $0.69 \pm 0.03$ |
| AE | $0.57 \pm 0.04$ | $0.60 \pm 0.04$ | $0.74 \pm 0.02$ |
| PCA | $0.70 \pm 0.02$ | $0.61 \pm 0.03$ | $0.57 \pm 0.03$ |
| ICA | $0.70 \pm 0.02$ | $0.58 \pm 0.03$ | $0.47 \pm 0.05$ |
| 333 parcels | | $0.55 \pm 0.03$ | |
| vSNR | dim = 2 | dim = 4 | dim = 32 |
| VAE ($\beta = 20$) | $0.61 \pm 0.29$ | $0.74 \pm 0.38$ | $\mathbf{0.81 \pm 0.50}$ |
| VAE ($\beta = 1$) | $0.58 \pm 0.31$ | $0.59 \pm 0.34$ | $0.73 \pm 0.36$ |
| AAE | $0.57 \pm 0.31$ | $0.56 \pm 0.30$ | $0.57 \pm 0.33$ |
| AE | $0.51 \pm 0.32$ | $0.56 \pm 0.35$ | $0.43 \pm 0.28$ |
| PCA | $0.57 \pm 0.38$ | $0.71 \pm 0.42$ | $0.76 \pm 0.45$ |
| ICA | $0.53 \pm 0.31$ | $0.71 \pm 0.41$ | $0.78 \pm 0.38$ |
| 333 parcels | | $0.74 \pm 0.40$ | |

Table 3: **Variability of parcel assignment within and between subjects using k-means clustering (k = 12 clusters).**

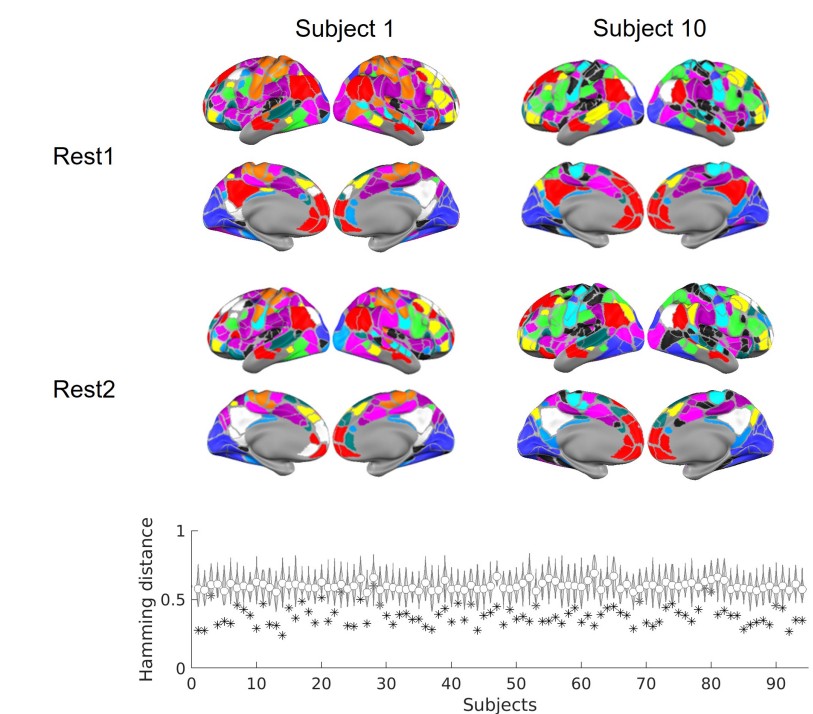

Figure 14: **Individual-specific network using k-means clustering with 12 networks (VAE ($\beta$ = 20), 2 dimensions).**

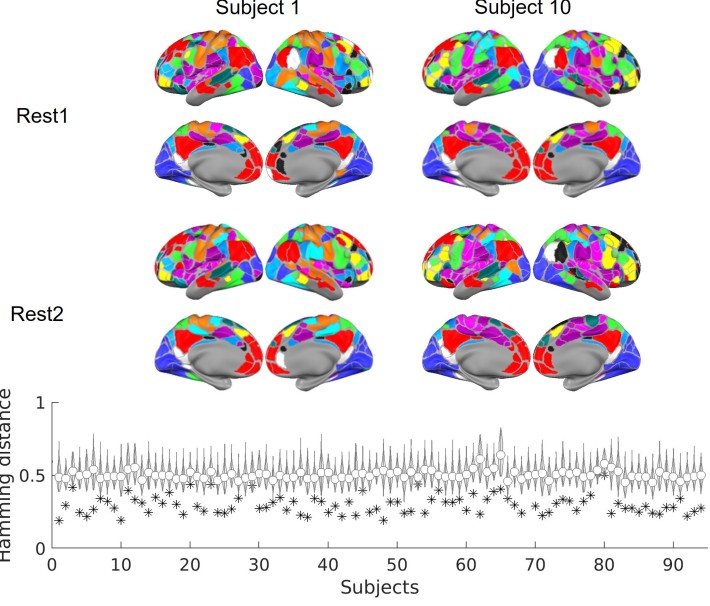

Figure 15: **Individual-specific network using k-means clustering with 12 networks (VAE ($\beta$ = 20), 4 dimensions).**

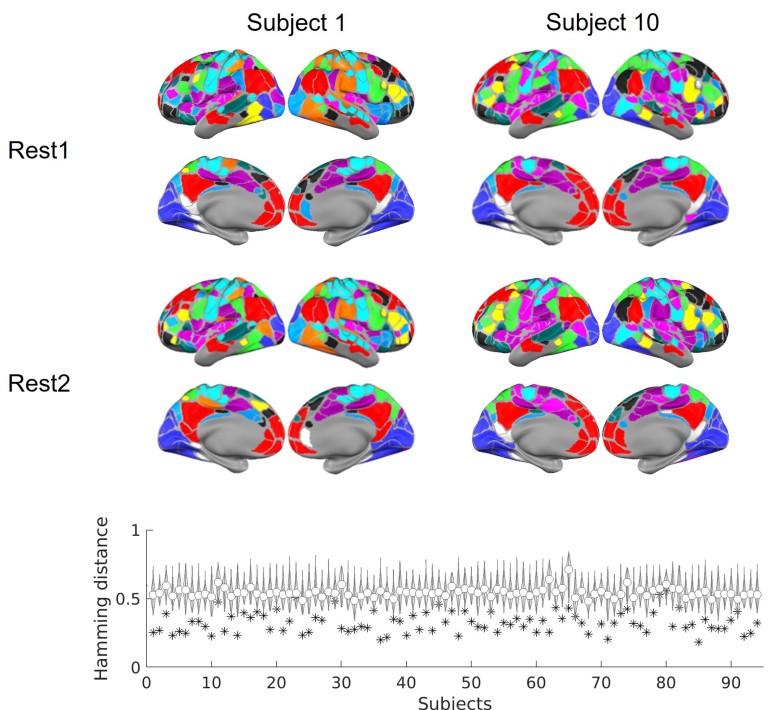

Figure 16: **Individual-specific network using k-means clustering with 12 networks (333 parcels).**

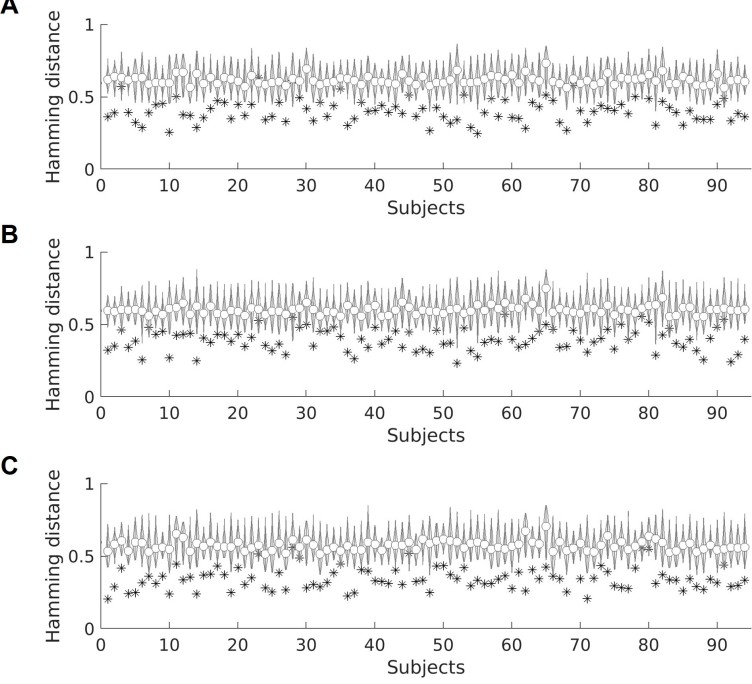

Figure 17: **Individual specificity of network assignment using k-means clustering with 12 networks (VAE ($\beta$ = 1)).** A) dimension = 2, B) dimension = 4, C) dimension = 32

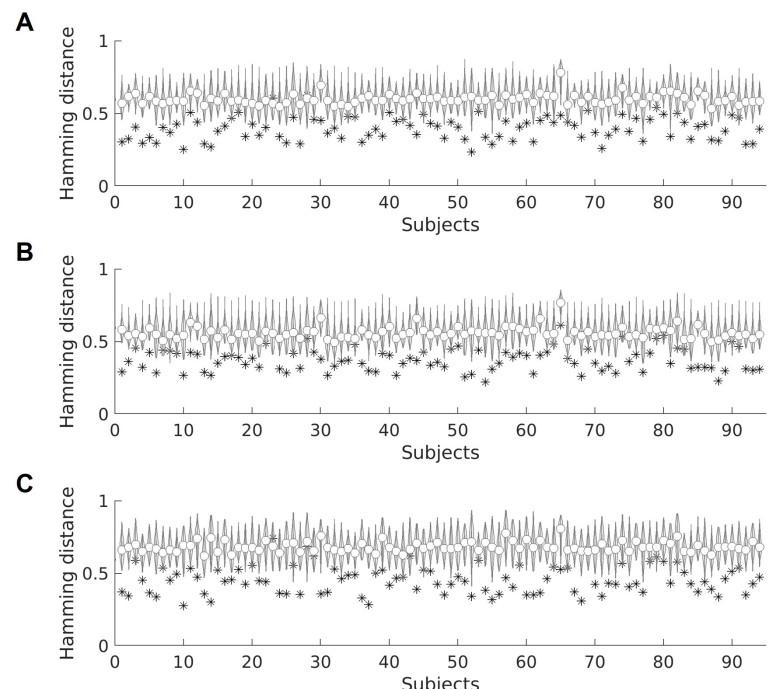

Figure 18: **Individual specificity of network assignment using k-means clustering with 12 networks (AAE).** A) dimension = 2, B) dimension = 4, C) dimension = 32

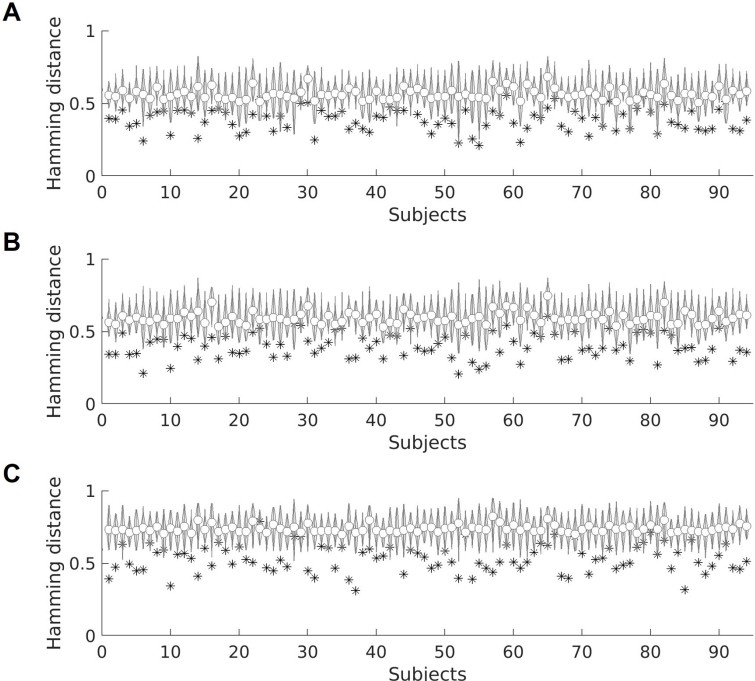

Figure 19: **Individual specificity of network assignment using k-means clustering with 12 networks (AE).** A) dimension = 2, B) dimension = 4, C) dimension = 32

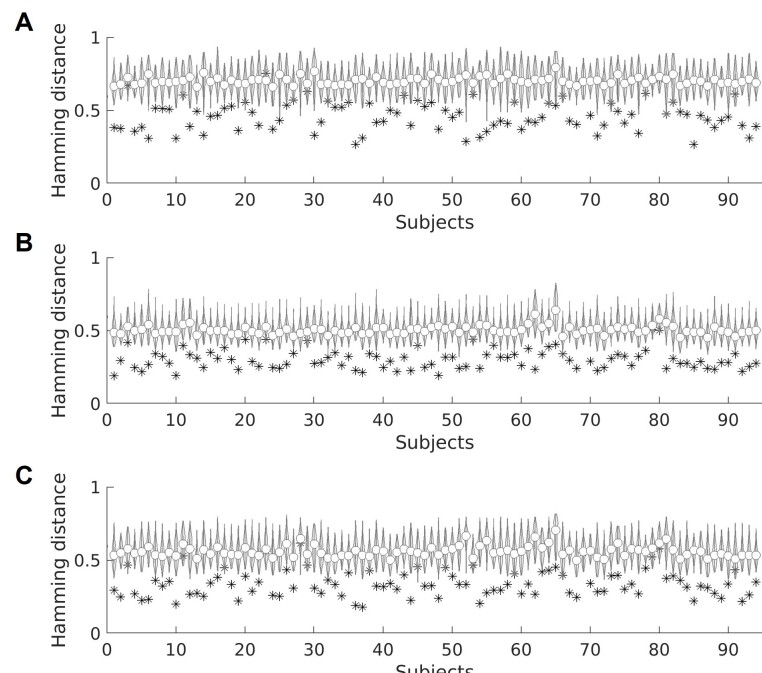

Figure 20: **Individual specificity of network assignment using k-means clustering with 12 networks (PCA).** A) dimension = 2, B) dimension = 4, C) dimension = 32

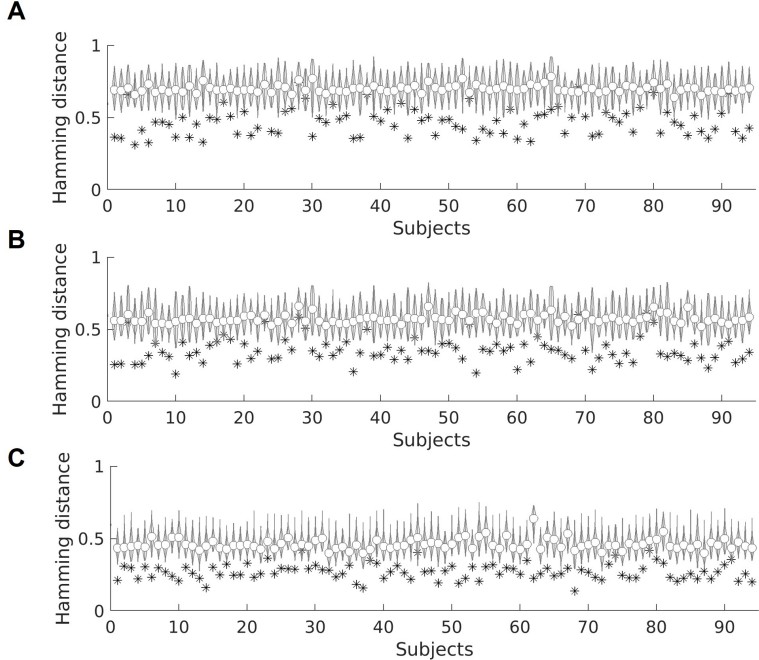

Figure 21: **Individual specificity of network assignment using k-means clustering with 12 networks (ICA).** A) dimension = 2, B) dimension = 4, C) dimension = 32

### A.13 IDENTIFICATION OF INDIVIDUALS USING THE FINGERPRINTING ANALYSIS

An alternative measure of individual specificity is the "fingerprinting" analysis (Finn et al., 2015). Since the parcel-wise connectome matrix (333 x 333) is symmetric, we concatenated the lower trian-

gle into a vector as a "barcode" for the first and second sessions, respectively. The similarity across sessions for the same and different subjects was calculated using Pearson's correlation between those "barcodes". For the latent representations with dimensions 2,4, and 32, we concatenated the values from all dimensions for each session as the "barcode" for the session and repeated the fingerprinting analysis. The average accuracy of identification was calculated as the average of two quantities: the proportion of correct identification of the subject from the session 1 data, and the proportion of correct identification from the session 2 data. We additionally calculated a Z-score which describes the relative magnitude of correlation between the sessions from the same subject and the sessions from different subjects. A Z-score was calculated for each subject as:

$$Z = \frac{r_{within} - Mean(r_{between})}{SD(r_{between})} \tag{5}$$

We found that both the accuracy (Figure 22A-C) and within-subject to between-subject contrast (Figure 22D-F) increased with the amount of data. Across all dimensions of 2,4 and 32 using VAE, PCA, and ICA, the fingerprinting accuracy in latent spaces was similar to that in the parcel space, especially for a long scan time ($> 20$ min). Overall, the difference in accuracy is larger with a shorter scan time, with the accuracy from the VAE ranked the highest among the dimensionality reduction methods with a marginal difference. Since incorrect identification would occur when even if only one other subject had a similar connectome (or connectome latents) to the subject in query, we also quantify the within-subject to between-subject contrast which is defined as the Z-score of the within-subject correlation normalized by the mean and standard deviation of the between-subject correlations. This contrast was higher with increasing latent dimensions and is higher for VAE compared to PCA and ICA, especially at higher latent dimensions (Figure 22D-F).

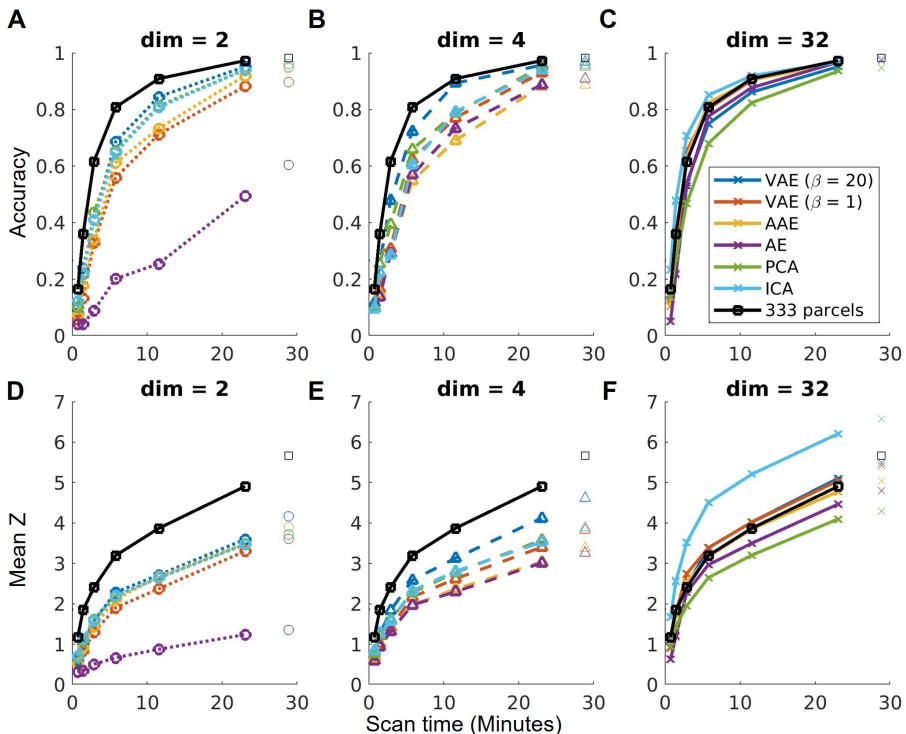

Figure 22: **Fingerprinting performance for 94 HCP subjects.** A-C) Fingerprinting accuracy. D-F) Mean Z-score of within-subject to between-subject contrast.

### A.14 PREDICTION OF BEHAVIORAL PHENOTYPE IN INDIVIDUALS

To demonstrate the ability of the embedded data to retain information about behavioral phenotypes in individual functional connectomes, we applied the learned dimensionality reduction mappings to a separate infant dataset (Baby Connectome Project(Howell et al., 2019)) with 301 sessions at 8-60

months. We predicted the sex and chronological age of each data sample in the parcel space (326 parcels with $326 \times (326\text{-}1)/2 = 52975$ unique features)(Tu et al., 2024) and the latent space ($326 \times$ zdim features, where zdim = 2, 4, 32) using AE-based or linear dimensionality reduction methods. We use a support vector machine regression for age and support vector machine classification for sex with $80\%$ of the data used for training and the remaining $20\%$ used in testing. We applied a ridge regularization where the hyper-parameter $\lambda$ was optimized with a 5-fold cross-validation approach from a range of 15 $\lambda$ values from $10^{-5}$ to $10^{-1}$ evenly distributed on a log scale. This process was repeated 1000 times by randomly splitting training and test data to generate an error bar. The age prediction performance was measured with correlation (r) and mean absolute error (MAE). The sex prediction performance was measured with accuracy and F1-score. We found that prediction performance generally increased with the number of dimensions, but the difference across methods was small except for when dimension = 2 (Figure 23).

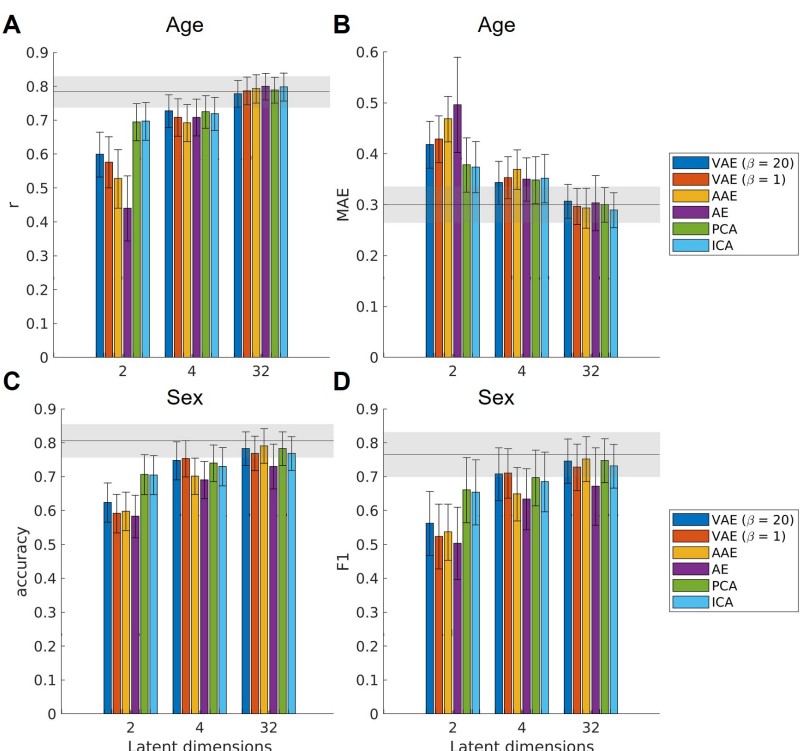

Figure 23: **Prediction of behavioral phenotype in individuals.** A) Age prediction correlation (r). B) Age prediction Mean absolute error (MAE). C) Sex prediction (accuracy). D) Sex prediction (F1-score). Error bars show mean and standard deviation across the 1000 samples. The horizontal line and shaded area show the performance (mean and standard deviation) using features in the parcel space.

### A.15 DIMENSIONALITY REDUCTION ON TIME SERIES DATA

In theory, one can run dimensionality reduction directly on spatiotemporal fMRI time series data, and then generate the seed map embeddings by correlating the original time series with this low-dimensional time series. However, the seed map correlations are likely less noisy than the raw time series because it is a summary metric. Latent embeddings obtained from dimensionality reduction directly on the seed maps could be easily projected back to original seed map images, which can then be overlaid with anatomy to provide intuitive visualizations and neuroscience insights (Figure 3). Also, doing dimensionality reduction directly on seed maps weighs each subject's data equally even when they have very different data acquisition lengths, and it is less likely that a few subjects with long acquisition in the training data would bias the embedding axes. Furthermore, the seed map correlations have constrained values ranging between -1 and 1 but the raw time series data can have different scales depending on the type of normalization. Direct seed map embedding can also

be applied to higher-level summary data such as the network-average functional connectivity across a group of subjects (Moore et al., 2024) so that individual seed maps can be juxtaposed with those network-average profiles to illustrate their positions relative to those network priors.

To empirically compare the dimensionality reduction on time series data and seed maps, we also conducted additional experiments using the time series data as the input to the model instead. We now have 27255 samples instead of the 594200 samples (5942 samples per subject x 100 subjects) in the training data. Similarly, the validation data samples decreased from 59420 to 3758. Overall, the reconstruction performance on the validation data is marginally lower when dimensionality reduction was performed on time series than on seed maps (Table 4). This difference in performance may be driven by the difference in the number of samples used to train the model.

| | Dimensionality reduction on time series | Dimensionality reduction on functional connectivity |
|---|---|---|
| VAE ($\beta = 20$) | | |
| dim = 2 | $0.57 \pm 0.05$ | $0.71 \pm 0.10$ |
| dim = 4 | $0.54 \pm 0.04$ | $0.74 \pm 0.08$ |
| dim = 32 | $0.51 \pm 0.01$ | $0.79 \pm 0.07$ |
| VAE ($\beta = 1$) | | |
| dim = 2 | $0.57 \pm 0.05$ | $0.71 \pm 0.09$ |
| dim = 4 | $0.54 \pm 0.04$ | $0.73 \pm 0.08$ |
| dim = 32 | $0.52 \pm 0.02$ | $0.83 \pm 0.06$ |
| AAE | | |
| dim = 2 | $0.57 \pm 0.06$ | $0.72 \pm 0.09$ |
| dim = 4 | $0.61 \pm 0.06$ | $0.73 \pm 0.08$ |
| dim = 32 | $0.73 \pm 0.05$ | $0.83 \pm 0.06$ |
| AE | | |
| dim = 2 | $0.57 \pm 0.05$ | $0.72 \pm 0.09$ |
| dim = 4 | $0.55 \pm 0.04$ | $0.73 \pm 0.07$ |
| dim = 32 | $0.52 \pm 0.02$ | $0.82 \pm 0.06$ |
| PCA | | |
| dim = 2 | $0.58 \pm 0.07$ | $0.67 \pm 0.10$ |
| dim = 4 | $0.63 \pm 0.07$ | $0.74 \pm 0.09$ |
| dim = 32 | $0.76 \pm 0.05$ | $0.84 \pm 0.06$ |
| ICA | | |
| dim = 2 | $0.58 \pm 0.07$ | $0.67 \pm 0.10$ |
| dim = 4 | $0.63 \pm 0.07$ | $0.74 \pm 0.09$ |
| dim = 32 | $0.76 \pm 0.05$ | $0.84 \pm 0.06$ |

Table 4: Reconstruction performance ($\eta^2$) on the validation data (10 subjects in WU120)

Next, we look at the embedded seed maps for 10 adult subjects from the test dataset of WU120 generated from 1) indirect embedding by embedding the time series ($2 \times T$) and then correlating with the parcel time series ($333 \times T$), and 2) direct embedding. We found that embedding the time series with linear methods produces seed map distributions that are similar to those obtained from directly embedding the seed maps (Figure 24). In addition, we repeated the behavioral phenotype prediction with the new seed map embeddings using the indirect method (a.k.a. embedding the time series first). The performance in predicting age and sex is similar between the direct and indirect embeddings (Figure 25).

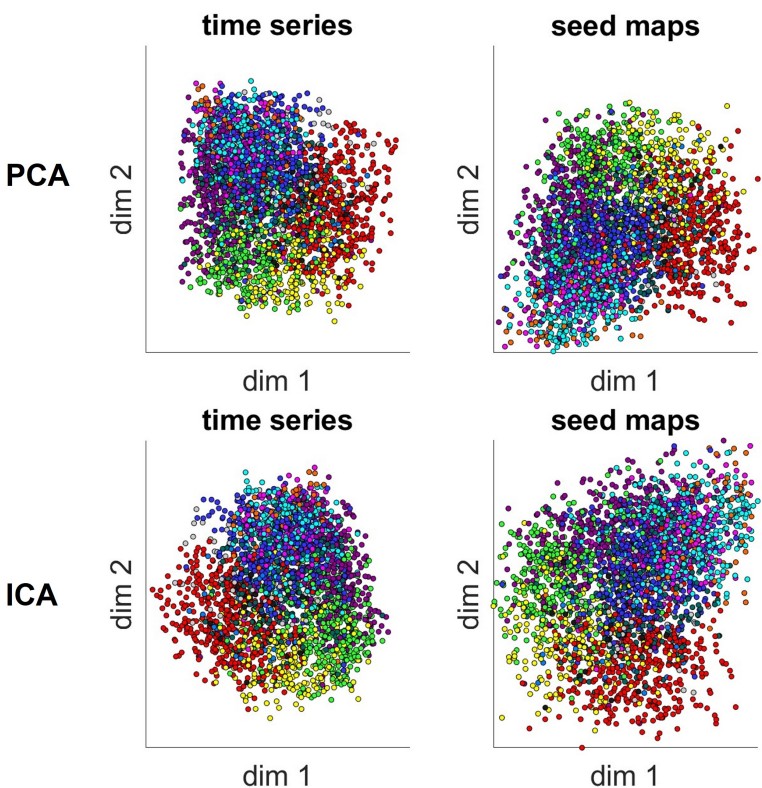

Figure 24: **Seed map latent embeddings using direct or indirect dimensionality reduction.** A) PCA on the time series data. B) PCA on the seed maps. C) ICA on the time series data. D) ICA on the seed maps. The colors correspond to the functional network labels in Figure 4A. The colors follow the network colors in Figure 4.

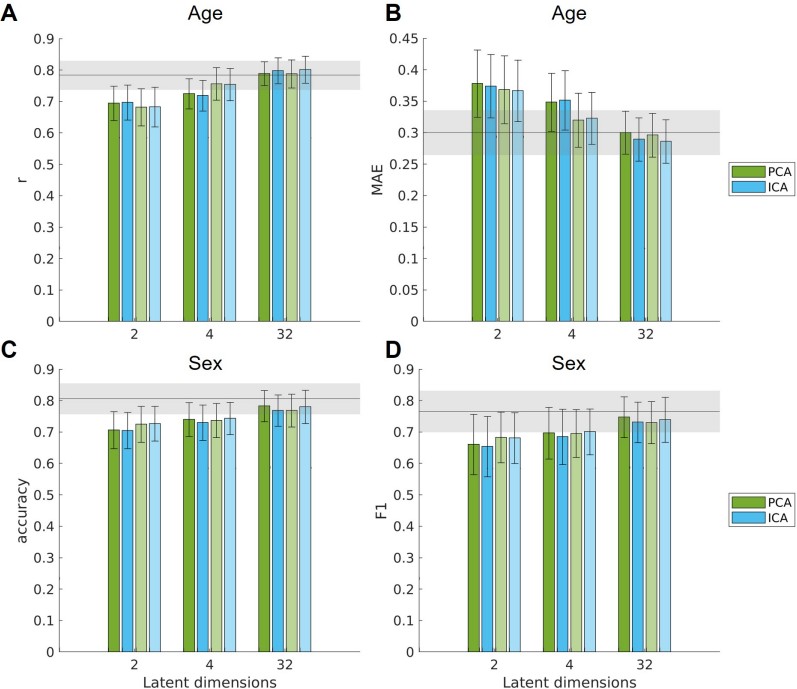

Figure 25: **Prediction of behavioral phenotypes using seed map embeddings generated directly or indirectly through dimensionality reduction on the time series.** The two darker bars on the left used direct seed map embedding and the two lighter bars on the right used indirect embedding through dimensionality reduction on the time series.

