# OpenReview forum: "Learning a Compact, Parcel-independent Representation of the fMRI Functional Connectivity"
_ICLR.cc/2025/Conference — Submitted to ICLR 2025_

### Official Review · Reviewer_9EKG · 2024-10-29

**Soundness:** 3
**Presentation:** 2
**Contribution:** 1
**Rating:** 5
**Confidence:** 3

**Summary:**

The authors train a variational autoencoder (VAE) to transform functional connectome seed maps into compressed latent representations that retain discriminatory power between individuals. Their hypothesis is that this approach improves on the standard practice of aggregating functional data into brain parcels which might leave room for further compression, while also collapsing meaningful vertex-level information. The authors conduct experiments that compute reconstruction accuracy,  and separation of subjects, comparing their VAE encoding to PCA, ICA, and a full parcellation encoding. They find that their approach separates subjects better than PCA and ICA, and clusters brain regions similarly to a established brain parcellations.

**Strengths:**

The goal of this work, dimensionality reduction and representation learning for functional connectomes, is a clear and a potentially impactful goal since functional surface maps are very high dimensional (tens of thousands of nodes). They also evaluate a variety of properties that would be desirable for feature learning including subject separation, agreement with existing parcellations, and reconstruction accuracy. Their method of latent feature embedding has an advantage over methods like t-SNE in that they can decode latent features into fMRI data. Their method is better than PCA and ICA at reconstruction accuracy and subject separation at 2 dimensions. The writing is generally clear and organized.

**Weaknesses:**

My largest concern is the significance of the contribution of this work. This paper cites a body of work by Kim et al. which also uses variational autoencoders to encode fMRI data with the same goals of reconstruction accuracy, and subject separation. In fact, figure 1 here is the same as Kim et al., 2021, just with a different seed map. Kim et al., 2021 also compares VAEs to PCA and ICA, further minimizing the novelty of this paper.

The authors cite computational complexity as a central motivation for the work. However, standard parcellations only use a couple hundred nodes, and datasets have around a hundred subjects. I am not convinced by their argument that computational complexity is a significant problem at this scale. After all, they cite a community detection algorithm which works on graphs over 1000x larger (Soman and Narang, 2011). While their method indeed is lower dimensional than parcellations with hundreds of areas, I would want to see complexity or runtime analysis if they claim that their method offers significant computational savings.

Minor presentation comments on Figure 3:
- I think the colormap for correlations is a bit unclear since some similar colors are in fact far away from each other (e.g. light/dark blue, yellow and light green). I suggest considering alternative "diverging" color maps.
- Given that there isn't much discussion of matching subjects across method, I suggest not using line plots in panels B-D and instead using a Strip plot, or box-and-whisker plots, or bar plots. I think those do a better job depicting differences.

**Questions:**

- Does figure 3 only show results for the test set?
- Are there other nonlinear feature learning methods that you can compare your method to? You rule-out t-SNE because the inverse mapping doesn't exist, but your comparisons (PCA, ICA) are both linear methods.

**Details Of Ethics Concerns:**

Figure 1 here is remarkably similar to Figure 1 in "Representation learning of resting state fMRI with variational autoencoder" by Kim et al., 2021. While this figure is a fairly generic description of a VAE, and Kim et al. is cited in the body text, the figure is not directly attributed, and some of the caption wording is identical. To me, this is a gray area so I am hoping to get a second opinion.

---

> ### Author Response · Authors · 2024-11-12
> **wrong paper**
>
> Hi,
>
> I believe this reviewer was looking at the wrong paper with a similar topic? The content mentions were not found in our paper and I don't have time series or Table 3... I also don't have ADNI/OASIS etc.

---

> > ### Comment · Reviewer_9EKG · 2024-11-12
> > **So Sorry!!**
> >
> > You are right - it looks like I swapped my reviews for two papers, I will try to rectify this ASAP.

---

> > > ### Comment · Reviewer_9EKG · 2024-11-12
> > > **Swapped reviews resolved**
> > >
> > > I edited the reviews so the problem should be resolved. Thanks for pointing it out, and apologies for the inconvenience.

---

> > > > ### Comment · Area_Chair_Dpp5 · 2024-11-12
> > > > **Happy to see an efficient interaction**
> > > >
> > > > Congratulations to both authors and review for swiftly identifying the problem and resolving it.

---

> ### Author Response · Authors · 2024-11-12
> **dual submission and novelty concern**
>
> Hi for your concern about dual submission and overlap with the Kim et al. paper, please see my responses below. I will reply to other concerns separately in a different comment (not sure if this is the right way to do it but it's the first time I'm submitting to a conference on OpenReview and I'm still learning).
>
> 1. The Kim et al. paper used fMRI time series data (temporal), and the current work uses the connectivity seed maps (spatial, correlation of time series) which tend to be "trait-like" fingerprints for subjects (one summary metric per session) and are much more commonly associated with traits/phenotype etc. unlike fMRI time series data. In addition, their training data and testing data were from the same dataset, and we want to demonstrate that the projections learned from one dataset (be it linear or nonlinear) could generate meaningful embeddings in a completely different dataset with very different acquisition protocols. Therefore, like atlases/parcellations, we could use a generic projection (whether it is linear or nonlinear) to obtain insights for/store new data instead of having a different embedding for each dataset (e.g. Margulies et al. 2016 PNAS and other fMRI dimensionality reduction). It was my original plan to further extend and demonstrate this in developmental datasets but I felt that would not fit in the limited pages in the paper.
>
> 2. We did adapt the figure from Kim et al. 2021 because we used the model architecture from the publically available code from Kim et al. 2021 https://github.com/libilab/rsfMRI-VAE as we stated in the methods. Because it seems like a generic model description and I could not find a good way to modify it while still retaining the meaning, I modified the figure. Is it possible for me to note "adapted from consent from Kim et al. 2021" to avoid plagiarism? Or is there any suggestion on how I could modify the figure but retain the meaning?

---

> > ### Comment · Reviewer_9EKG · 2024-11-13
> >
> > The figure should be clearly described (i.e. in the caption) as adapted from the original source: https://guides.himmelfarb.gwu.edu/APA/image-figure#:~:text=When%20you%20use%20a%20figure,or%20adapted%20for%20your%20paper.

---

> ### Author Response · Authors · 2024-11-27
>
> Dear reviewer,
>
> Thank you for your extensive review of our paper. I would like to respond to some of your major concerns below. I understand that this might not be your favorite paper, but I would greatly appreciate it if you could provide me with feedback on whether I have understood and addressed your concerns and what additional areas for improvement are needed. This is my first submission to ICLR, and I would like your honest opinion on whether a resubmission to the same or a similar venue with some revisions of the original paper is appropriate, or if you suggest other venues that might be a better fit for the content. This is also my first submission of a paper on Open Review, so please excuse me and point out potential improvements for my response formatting.
>
> ## Significance of Contribution
> We have fixed the potential copyright violation issue by changing the captions in Figure 1 to correctly identify and acknowledge the potential sources. We appreciate your careful examination of the references in our paper.
>
> Like you suggested, initially we did intend to apply the pre-trained model provided by [1] for our purpose to project new data with different parcel definitions onto the same space, with the expectation that the two kinds of data are highly related. However, potentially due to the differences in the data scale (fMRI activity) versus functional connectivity (with a bound between 1 and -1) and/or the data processing differences, the reconstruction performance was not very good. In addition to using temporal time series data in [1], that paper did not demonstrate generalizability to a different dataset and settled on a very large latent dimension (256) that is almost comparable with the parcel space. Here we explored the model performance at much lower numbers of latent dimensions.
>
> Latent embeddings obtained from dimensionality reduction directly on the seed maps rather than time series [1] could be easily projected back to original seed map images, which can then be overlaid with anatomy to provide intuitive visualizations and neuroscience insights (Figure 3). Also, doing dimensionality reduction directly on seed maps weighs each subject's data equally even when they have very different data acquisition lengths and it is less likely that a few subjects with long acquisition in the training data would bias the embedding axes. Furthermore, the seed map correlations have constrained values ranging between -1 and 1 but the raw time series data can have different scales based on normalization. Direct seed map embedding can be applied to higher-level summary data such as the network-average functional connectivity across a group of subjects (network templates) [2].
>
> [1] Kim, J.-H., Zhang, Y., Han, K., Wen, Z., Choi, M., Liu, Z., 2021. Representation learning of resting state fMRI with variational autoencoder. NeuroImage 241, 118423. https://doi.org/10.1016/j.neuroimage.2021.118423
>
> [2] Moore, L.A., Hermosillo, R.J.M., Feczko, E., Moser, J., Koirala, S., Allen, M.C., Buss, C., Conan, G., Juliano, A.C., Marr, M., Miranda-Dominguez, O., Mooney, M., Myers, M., Rasmussen, J., Rogers, C.E., Smyser, C.D., Snider, K., Sylvester, C., Thomas, E., Fair, D.A., Graham, A.M., 2024. Towards personalized precision functional mapping in infancy. Imaging Neuroscience 2, 1–20. https://doi.org/10.1162/imag_a_00165

---

> ### Author Response · Authors · 2024-11-27
> **Response to reviewer 9EKG (part 2)**
>
> ## Motivation for improved computational efficiency
> You are correct to some extend the computational complexity of the parcel space is not too bad for hundreds of subject/sessions. While the number of nodes (100-1000) and edges (4950-499500) might not seem big for an individual functional connectivity matrix, the total data space can become large when the "layers" in multi-layer community detection include all individuals (e.g., for UK Biobank it would be 40000+ [3]), longitudinal sessions, and time windows within a session [4-6]. In addition, recently people combine across multiple datasets for life span studies [7] which can increase the sample size further.
>
> We conducted additional analysis (both theoretical and experimental) on the run-time of multi-layer modularity maximization [8-9] in the appendix section A2. Overall, we confirmed that the computational complexity is about O(n(log(n)) where n is the number of nodes in each layer. We acknowledge that this improvement in computational efficiency is useful but not the main motivation that should drive the study. We have changed the wording in introduction to reflect that.
>
> [3] Horien, C., Noble, S., Greene, A.S., Lee, K., Barron, D.S., Gao, S., O’Connor, D., Salehi, M., Dadashkarimi, J., Shen, X., Lake, E.M.R., Constable, R.T., Scheinost, D., 2021. A hitchhiker′s guide to working with large, open-source neuroimaging datasets. Nat Hum Behav 5, 185–193. https://doi.org/10.1038/s41562-020-01005-4
>
> [4] de Domenico, M., 2017. Multilayer modeling and analysis of human brain networks. GigaScience 6, 1. https://doi.org/10.1093/gigascience/gix004
>
> [5] Muldoon, S.F., Bassett, D.S., 2016. Network and Multilayer Network Approaches to Understanding Human Brain Dynamics. Philosophy of Science 83, 710–720. https://doi.org/10.1086/687857
>
> [6] Betzel, R.F., Bertolero, M.A., Gordon, E.M., Gratton, C., Dosenbach, N.U.F., Bassett, D.S., 2019. The community structure of functional brain networks exhibits scale-specific patterns of inter- and intra-subject variability. NeuroImage 202. https://doi.org/10.1016/j.neuroimage.2019.07.003
>
> [7]Sun, L., Zhao, T., Liang, X., Xia, M., Li, Q., Liao, X., Gong, G., Wang, Q., Pang, C., Yu, Q., Bi, Y., Chen, P., Chen, R., Chen, Y., Chen, T., Cheng, J., Cheng, Y., Cui, Z., Dai, Z., Deng, Y., Ding, Y., Dong, Q., Duan, D., Gao, J.-H., Gong, Q., Han, Y., Han, Z., Huang, C.-C., Huang, R., Huo, R., Li, L., Lin, C.-P., Lin, Q., Liu, B., Liu, C., Liu, N., Liu, Ying, Liu, Yong, Lu, J., Ma, L., Men, W., Qin, S., Qiu, J., Qiu, S., Si, T., Tan, S., Tang, Y., Tao, S., Wang, D., Wang, F., Wang, J., Wang, P., Wang, X., Wang, Y., Wei, D., Wu, Y., Xie, P., Xu, X., Xu, Y., Xu, Z., Yang, L., Yuan, H., Zeng, Z., Zhang, H., Zhang, X., Zhao, G., Zheng, Y., Zhong, S., Alzheimer’s Disease Neuroimaging Initiative, C.-C., He, Y., 2023. Functional connectome through the human life span. https://doi.org/10.1101/2023.09.12.557193
>
> [8] Mucha, P.J., Richardson, T., Macon, K., Porter, M.A., Onnela, J.-P., 2010. Community Structure in Time-Dependent, Multiscale, and Multiplex Networks. Science 328, 876–878. https://doi.org/10.1126/science.1184819
>
> [9] A generalized Louvain method for community detection implemented in MATLAB," https://github.com/GenLouvain/GenLouvain (2011-2019).

---

> ### Author Response · Authors · 2024-11-27
> **Response to reviewer 9EKG (part 3)**
>
> ## Figure 3 presentation
> > Does figure 3 only show results for the test set?
>
> Yes. Figure 3 only showed results for the 10 test subjects (WU120, same acquisition, and processing as the training data) and the 94 subjects (2 sessions each) from HCP.
>
> > I think the colormap for correlations is a bit unclear since some similar colors are in fact far away from each other (e.g. light/dark blue, yellow,w and light green). I suggest considering alternative "diverging" color maps.
>
> We considered that, but since this color map has a cooler color for negative and a warmer color for positive with darker color meaning closer to zero, we think that it still makes sense. Also, this colormap in the Connectome Workbench software was used in a lot of HCP-related studies and would be more resonating with readers familiar with that literature [10-12].
>
> > Given that there isn't much discussion of matching subjects across method, I suggest not using line plots in panels B-D and instead using a Strip plot, or box-and-whisker plots, or bar plots. I think those do a better job depicting differences.
>
> Each line represents the same subject across different methods so I kept the lines. However, I took your feedback and added some jittering in the scatterplot so it's clearer to see the individual datapoints.
>
> [10] Glasser, M.F., Coalson, T.S., Bijsterbosch, J.D., Harrison, S.J., Harms, M.P., Anticevic, A., Van Essen, D.C., Smith, S.M., 2018. Using temporal ICA to selectively remove global noise while preserving global signal in functional MRI data. NeuroImage 181, 692–717. https://doi.org/10.1016/j.neuroimage.2018.04.076
> [11] Glasser, M.F., Coalson, T.S., Robinson, E.C., Hacker, C.D., Harwell, J., Yacoub, E., Ugurbil, K., Andersson, J., Beckmann, C.F., Jenkinson, M., Smith, S.M., Van Essen, D.C., 2016. A multi-modal parcellation of human cerebral cortex. Nature 536, 171–178. https://doi.org/10.1038/nature18933
> [12] Glasser, M.F., Sotiropoulos, S.N., Wilson, J.A., Coalson, T.S., Fischl, B., Andersson, J.L., Xu, J., Jbabdi, S., Webster, M., Polimeni, J.R., Van Essen, D.C., Jenkinson, M., 2013. The minimal preprocessing pipelines for the Human Connectome Project. NeuroImage, Mapping the Connectome 80, 105–124. https://doi.org/10.1016/j.neuroimage.2013.04.127

---

> ### Author Response · Authors · 2024-11-27
> **Response to reviewer 9EKG (part 4)**
>
> ## Comparison to other nonlinear feature learning methods
> > Are there other nonlinear feature learning methods that you can compare your method to? You rule-out t-SNE because the inverse mapping doesn't exist, but your comparisons (PCA, ICA) are both linear methods.
>
> We recognize that it is a significant weakness to only compare to linear methods. Therefore, we added the conventional autoencoder and generative adversarial network (GAN)-based adversarial autoencoder [13]. The results were updated in the revised manuscript (Figures 2-4).
>
> [13] Makhzani, A., Shlens, J., Jaitly, N., Goodfellow, I., Frey, B., 2016. Adversarial Autoencoders. ICLR. https://doi.org/10.48550/arXiv.1511.05644

---

> > ### Comment · Reviewer_9EKG · 2024-12-02
> >
> > I think the authors improved the paper by adding other nonlinear methods, and adding an argument about the potential usefulness of a lower dimensional representation of functional networks. However, the main result of this paper seems to still be that autoencoders can compress networks into 2 dimensions with slightly less reconstruction error than linear methods. I think the paper is still missing a clear demonstration of why this will change the field of fMRI analysis. It is not clear whether 2 vs. 4 dimensional representation is the critical difference between analysis algorithms being able to run or not. I am excited to raise my review score based on the recent improvement, but I am sorry to say that I still cannot recommend this work for publication.

---

> > > ### Author Response · Authors · 2024-12-02
> > > **Thank you for your feedback**
> > >
> > > Dear reviewer,
> > >
> > > Thanks for reading our revision. It might be due to our presentation, but the main contribution is not that the VAE had lower reconstruction error at two dimensions, but the ability to reuse precomputed mapping from an independent dataset to align FC from different parcels from different subjects to provide useful insight, especially with the VAE (at 2-dimension it aids visualization and separates the functional networks better in Figure 4). As far as we know, the existing analyses tend to do dimensionality reduction on each dataset (and mostly on the spatiotemporal domain) which is both computationally costly and doesn't compare to new data straightforwardly. We will think about what kind of analysis and presentation would make this more explicit in the next submission.

---

### Official Review · Reviewer_xAoY · 2024-11-01

**Soundness:** 2
**Presentation:** 3
**Contribution:** 2
**Rating:** 3
**Confidence:** 4

**Summary:**

This paper is based on the hypothesis that ‘it is possible to learn a compact representation of functional connectivity data to enhance computational efficiency without compromising the essential structure of the original data’. This is achieved by projecting high-dimensional fMRI data onto a common low-dimensional latent space through a variational autoencoder, the study aims to reduce redundancy and improve cross-individual comparability, especially when different parcellation schemes are used. Using different performance metrics, the study explores how well key features in the original connectome are preserved across various dimensionality reduction methods and dimensions. The findings highlight that a variational autoencoder performs better than linear methods at lower dimensions and suggest that low-dimensional representations can enhance reproducibility and support open science.

**Strengths:**

1. The main motivation of this study was clearly explained. This study is based on the hypothesis that ‘it is possible to learn a compact representation of functional connectivity data to enhance computational efficiency without compromising the essential structure of the original data’. It is reasonable in the neuroscience field.
2. The proposed framework and geometric reformatting were clearly explained. The author applies the VAE framework to extract the low-dimensional representation of the reformatting image.

**Weaknesses:**

1. Although the proposed framework and geometric reformatting were clearly explained, however, this proposed framework lacks innovation, as the use of VAEs in neuroimaging is already well-established.
2. Dimensionality reduction and data compression to improve cross-individual comparability and computational efficiency are common strategies in many studies.
3. From my point of view, the use of an autoencoder like VAE inherently involves a tradeoff between interpretability and data embedding. While the VAE effectively embeds high-dimensional data into a compact latent space, the representation in a new state space lacks intuitive interpretability. How about the comparison of other dimensional reduction methods, that directly extract the spatial or temporal modes in the original state space?

**Questions:**

1. What unique advantages does this framework offer in terms of cross-individual comparability and computational efficiency over other commonly used dimensionality reduction techniques, like PCA or ICA?
2. It is a tradeoff between interpretability and data compactness. In the past 2-6 years, it has been a popular topic in neuroscience that extracts harmonic modes/representations from structural networks or functional networks.
3. Typically, the estimation of functional connectivity might lose temporal information of neural data, how about the direct embedding of temporal signal in the proposed framework?

---

> ### Author Response · Authors · 2024-11-26
> **xAoY**
>
> Dear reviewer,
>
> Thank you for your extensive review of our paper. I would like to respond to some of your major concerns below. I understand that this might not be your favorite paper, but I would greatly appreciate it if you could provide me with feedback on whether I have understood and addressed your concerns and what additional areas for improvement are needed. This is my first submission to ICLR, and I would like your honest opinion on whether a resubmission to the same or a similar venue with some revisions of the original paper is appropriate, or if you suggest other venues that might be a better fit for the content. This is also my first submission of a paper on Open Review, so please excuse me and point out potential improvements for my response formatting.
>
> > 1.	What unique advantages does this framework offer in terms of cross-individual comparability and computational efficiency over other commonly used dimensionality reduction techniques, like PCA or ICA?
>
> We compared the linear dimensionality reduction methods (PCA,ICA) and autoencoder-based dimensionality reduction methods. Indeed, we found that the results in a lot of aspects are similar and sometimes the linear methods may perform better (e.g. reconstruction at 32 dimensions). However, the autoencoder-based methods had an advantage in separating the finer details of the functional networks in a 2-dimensional representation (Figure 2 and 4). In addition, in terms of preservation of biologically meaningful structures including the separation into functional networks and consistency of interindividual variability across sessions, the VAE with $\beta$ = 20 seems to perform better. If by "commonly used dimensionality reduction techniques" you meant dimensionality reduction in the fMRI time series data, please refer to the Appendix section A15: Dimensionality reduction on time series data.
>
> > 2.	It is a tradeoff between interpretability and data compactness. In the past 2-6 years, it has been a popular topic in neuroscience that extracts harmonic modes/representations from structural networks or functional networks.
>
> While the $\beta$ variational autoencoder is effective in disentangling generative factors in images (e.g. faces)[1], our approach indeed lacks interpretability and is a descriptive/phenomenological model rather than a mechanistic model. We did not appreciate the structural-functional interplay like the popular methods exploring harmonic modes/eigen modes in the brain from anatomy [2-5]. Despite that, we believe that it still provides a useful visualization and utility to juxtapose functional connectivity seed maps from different parcel definitions. We have acknowledged this limitation in the discussion.
>
> [1] Higgins, I., Matthey, L., Pal, A., Burgess, C., Glorot, X., Botvinick, M., Mohamed, S., Lerchner, A., 2017. β-VAE: LEARNING BASIC VISUAL CONCEPTS WITH A CONSTRAINED VARIATIONAL FRAMEWORK. Presented at the ICLR.
>
> [2] Pang, J.C., Aquino, K.M., Oldehinkel, M., Robinson, P.A., Fulcher, B.D., Breakspear, M., Fornito, A., 2023. Geometric constraints on human brain function. Nature 618, 566–574. https://doi.org/10.1038/s41586-023-06098-1
>
> [3] Atasoy, S., Deco, G., Kringelbach, M.L., Pearson, J., 2018. Harmonic Brain Modes: A Unifying Framework for Linking Space and Time in Brain Dynamics. Neuroscientist 24, 277–293. https://doi.org/10.1177/1073858417728032
>
> [4] Atasoy, S., Roseman, L., Kaelen, M., Kringelbach, M.L., Deco, G., Carhart-Harris, R.L., 2017. Connectome-harmonic decomposition of human brain activity reveals dynamical repertoire re-organization under LSD. Sci Rep 7, 17661. https://doi.org/10.1038/s41598-017-17546-0
>
> [5] Atasoy, S., Donnelly, I., Pearson, J., 2016. Human brain networks function in connectome-specific harmonic waves. Nat Commun 7, 10340. https://doi.org/10.1038/ncomms10340

---

> ### Author Response · Authors · 2024-11-26
> **Response to reviewer xAoY (part 2)**
>
> > 3.	Typically, the estimation of functional connectivity might lose temporal information of neural data, how about the direct embedding of temporal signal in the proposed framework?
>
> It is true that the estimation of functional connectivity might lose temporal information of neural data. While we are more interested in the trait-like brain connectivity signature across subjects/across development so the loss of temporal information is not a concern, we acknowledged this as a limitation in our discussion.
>
> In theory, one can run dimensionality reduction directly on spatiotemporal fMRI time series data, and then generate the seed map embeddings by correlating the original time series with this low-dimensional time series. However, the seed map correlations are likely less noisy than the raw time series because it is a summary metric. Latent embeddings obtained from dimensionality reduction directly on the seed maps could be easily projected back to original seed map images, which can then be overlaid with anatomy to provide intuitive visualizations and neuroscience insights (Figure 3). Also, doing dimensionality reduction directly on seed maps weighs each subject's data equally even when they have very different data acquisition lengths and it is less likely that a few subjects with long acquisition in the training data would bias the embedding axes. Furthermore, the seed map correlations have constrained values ranging between -1 and 1 but the raw time series data can have different scales based on normalization. Direct seed map embedding can be applied to higher-level summary data such as the network-average functional connectivity across a group of subjects in network templates [6].
>
> We have conducted additional analyses (direct embedding of temporal signal) and included the relevant results in the Appendix A.15: dimensionality reduction on time series data.
>
> [6] Moore, L.A., Hermosillo, R.J.M., Feczko, E., Moser, J., Koirala, S., Allen, M.C., Buss, C., Conan, G., Juliano, A.C., Marr, M., Miranda-Dominguez, O., Mooney, M., Myers, M., Rasmussen, J., Rogers, C.E., Smyser, C.D., Snider, K., Sylvester, C., Thomas, E., Fair, D.A., Graham, A.M., 2024. Towards personalized precision functional mapping in infancy. Imaging Neuroscience 2, 1–20. https://doi.org/10.1162/imag_a_00165

---

> > ### Comment · Reviewer_xAoY · 2024-11-30
> >
> > Thanks for your detailed response. The main idea of this paper is indeed interesting, but it applies an existing classical framework for scientific exploration. I believe it may be more suitable for some neuroscience-related journals. For example, nature neuroscience, nature methods, neuroimage or imaging neuroscience.

---

> > > ### Author Response · Authors · 2024-11-30
> > > **Thank you for your comment**
> > >
> > > Dear reviewer,
> > >
> > > Thank you for your comment. We appreciate you taking the time to read and review our paper.

---

> > > > ### Comment · Reviewer_xAoY · 2024-12-03
> > > >
> > > > I have carefully reviewed the articles you cited, and over 80% of them are neuroscience-related journals. Your study is interesting. Perhaps submitting to a neuroscience-related journal would be more appropriate. I hope your study contributes to everyone in the field of neuroscience.

---

### Official Review · Reviewer_GbuD · 2024-11-01

**Soundness:** 3
**Presentation:** 3
**Contribution:** 2
**Rating:** 5
**Confidence:** 4

**Summary:**

The approach explores a beta-VAE for the compression of functional neuroimaging data by use of a spherical projection of the cortical surface evenly sampled to form an image that is used by a standard CNN based encoder-decoder variational autoencoder framework with a Gaussian prior as distribution of the variational bottleneck. The image used as input is derived from a seed map defining the Pearson correlation from the seed region to all other regions. The approach is contrasted toICA and PCA and performance quantified in terms of reconstruction ability from the compressed representation, silhouette index to quantify separation of networks in terms of predefined functional regions, as well as ratio of intra to intersubject variability as a proxy of reliability (i.e. measurements of same subject should have similar embeddings as opposed to the embeddings of different subjects) considering data from the human connectome project (HCP).

**Strengths:**

The paper is very well written and clear in its presentation.

The experimentation is carefully executed and includes quite a bit of additional investigations also in the supplementary. The evaluation criteria are sound (but could be strengthened, see weaknesses).

The use of seed based maps are interesting and can potentially have merits in general providing robust representations of functional neuroimaging data.

**Weaknesses:**

The approach is not contrasted any conventional modeling of the same data.

The approach is rather straightforward using a conventional \beta-VAE methodologically and the key contribution here is the use of seed maps rather than operating on the raw spatio-temporal data.

The approach is using qualitative evaluation approaches, i.e. quantitatively evaluating reconstruction, homogeneity by SI and same subject invariance when compared to different subjects are interesting metrics, but not necessarily of strong neuroscientific impact. I.e. a poor model focusing on noisy signals may have high subject consistency as noise/bias may be subject specific, produce high degree of homogeneity and lend itself well reconstructed as such noise confounders may be prominent as fMRI generally suffers from poor SNRs. As such, the methodology is not compared to ground truth information of neuroscientific interest such as recovery of task responses in task data, ability to predict properties of the individuals such as age, gender and cognitive capabilities etc (and why I deem the results qualitative). Such data is available from the HCP cohort and would strengthen the study to include and see what has been learned in regards to neuro- and cognitive-science relevant aspects.

The results are not so convincing. It seems simple methods such as ICA and PCA when applied with larger dimensions provide better reconstruction quality than the beta-VAE as indicated by the results of Figure 3. I find this somewhat surprising as I would expect the beta-VAE to be able through its non-linear modeling to efficiently compress the signal characteristics of the seed maps. This would be good to further elaborate upon as it then becomes unclear why to use advanced modeling approaches as opposed to very simple procedures such as ICA and PCA.

I understand that given the metrics and the uniqueness of considering seed-derived maps there are no natural alternative modeling procedures to consider. However, I would have liked to see conventional PCA and ICA compression on the time series and their reconstructions of seeds to further understand if this type of information cannot already be reproduced in such conventional neuroimaging analyses.

Also, I think the paper would substantially improve to consider prediction of external information such as demography and cognitive abilities available in the HCP cohort to ground the methodology’s utility more quantitatively in terms of such ground truth information available for the individuals. It would in this context also be possible to understand and compare the proposed seed-based beta-VAE compressions utility when compared to standard neuroimaging compression methodologies operating directly on the spatio-temporal data for which there is a large literature using various approaches to predict aspects of the individuals of neuroscientific interest.

**Questions:**

How would PCA and ICA compare when applied to the spatio-temporal data as opposed to seed maps, i.e. filtering the data and reconstruction seed based maps from the filtered representations?

Why is \beta-VAE inferior in reconstructing the seed maps when using more dimensions when compared to ICA and PCA, and how would you generally tune for \beta in the \beta-VAE?

Why does ICA and PCA differ when they are reconstructing the same subspaces – this is unclear to me, please clarify, i.e. ICA is typically just a rotation of the corresponding PCA space – I believe I am missing an understanding on how these two approaches in this context become different.

Can you evaluate performance on information available at the subject level such as demography and cognitive capabilities in the HCP cohort based on the compressed representation?

- and how would such analysis compare to current SOTA supervised and representation learning approaches applied to fMRI in such tasks?

---

> ### Author Response · Authors · 2024-11-26
> **Response to reviewer GbuD**
>
> Dear reviewer,
>
> Thank you for your extensive review of our paper. I would like to respond to some of your major concerns below. I understand that this might not be your favorite paper, but I would greatly appreciate it if you could provide me with feedback on whether I have understood and addressed your concerns and what additional areas for improvement are needed. This is my first submission to ICLR, and I would like your honest opinion on whether a resubmission to the same or a similar venue with some revisions of the original paper is appropriate, or if you suggest other venues that might be a better fit for the content. This is also my first submission of a paper on Open Review, so please excuse me and point out potential improvements for my response formatting.
>
> # Questions about PCA and ICA
> > How would PCA and ICA compare when applied to the spatio-temporal data as opposed to seed maps, i.e. filtering the data and reconstruction seed based maps from the filtered representations?
>
> We have conducted additional experiments where we applied PCA and ICA directly to the spatiotemporal data and reconstructed the seed-based maps from the embedded spatiotemporal data. The comparative results are provided in the appendix section A15: Dimensionality reduction on time series data.
>
> > Why is \beta-VAE inferior in reconstructing the seed maps when using more dimensions when compared to ICA and PCA, and how would you generally tune for \beta in the \beta-VAE?
>
> Our suspicion is that the functional connectivity data has a predominantly linear structure and most of the variances can be captured in a few dimensions [1]. PCA is relatively unconstrained whereas the variational autoencoder (especially when $\beta$>1) constrained the latent distribution to approximate a Gaussian distribution. Other evidence where a constrained harmonic mode model produces inferior reconstruction performance at some number of dimensions exists in the literature as well. https://www.nature.com/articles/s41586-023-06098-1/figures/9. Another possibility is that when given too many latent variables, the variational autoencoder picks up the idiosyncracies in the training data which makes it perform a worse reconstruction in the test data. The tuning of the beta is described Appendix A.6 where we find the sweet spot based on KL divergence and reconstruction error.
>
> [1] Margulies, D.S., Ghosh, S.S., Goulas, A., Falkiewicz, M., Huntenburg, J.M., Langs, G., Bezgin, G., Eickhoff, S.B., Castellanos, F.X., Petrides, M., Jefferies, E., Smallwood, J., 2016. Situating the default-mode network along a principal gradient of macroscale cortical organization. Proc. Natl. Acad. Sci. U.S.A. 113, 12574–12579. https://doi.org/10.1073/pnas.1608282113
>
> [2] Pang, J.C., Aquino, K.M., Oldehinkel, M., Robinson, P.A., Fulcher, B.D., Breakspear, M., Fornito, A., 2023. Geometric constraints on human brain function. Nature 618, 566–574. https://doi.org/10.1038/s41586-023-06098-1
>
> > Why does ICA and PCA differ when they are reconstructing the same subspaces – this is unclear to me, please clarify, i.e. ICA is typically just a rotation of the corresponding PCA space – I believe I am missing an understanding on how these two approaches in this context become different.
>
> I apologize for not being able to understand your question. Is this referring to Figure 4H and 4K or something else? Your intuition is correct, in most of the metrics PCA and ICA are very similar/near identical.

---

> > ### Author Response · Authors · 2024-11-26
> > **Response to reviewer GbuD (part2)**
> >
> > ## Comparison to Related Work in Literature (novelty concerns)
> > > How would such analysis compare to current SOTA supervised and representation learning approaches applied to fMRI in such tasks?
> >
> > We agree that a discussion with prior work is crucial to demonstrate the contribution for the work. We now added a detailed discussion in the context of previous work in Section 2: Related work in the revised draft.
> >
> > While a lot of human neuroimaging work uses deep learning to capture fMRI activity, structural connectivity and functional connectivity, many of the work (including the ones sited by you) were using abstract nodes in either independent components or predefined regions, and have one single goal, e.g. to make predictions (on disease classification, prediction of masked brain activity etc.). Moreover, many of them consider the connectome matrix from one subject as a single sample, while our data samples are seed maps with thousands of samples in each subject. These were very different from our goal of obtaining a general, compact latent space for comparing data from different parcels (can be predefined regions or independent components above) with **pre-computed mappings**. Most of the existing work also focused on a specific machine learning benchmark within a single dataset instead of obtaining low-dimensional embeddings and test different benchmarks on multiple datasets. Therefore, we find it hard to directly compare our performance to those models because we could not identify one model that is directly comparable, unlike the other computer vision application with a standard task and metric (e.g., classification with MNIST or CIFAR-10 dataset with different variations of autoencoders).
> >
> > The most similar work in spirit is the representing of functional connectivity data into principal gradients to visualize the dominant spatial modes in functional connectivity [7-8] , but the samples in the latent distribution could not be “back-projected” to the original space to provide an intuitive visualization of the effect of varying the gradient from one end to another on the appearance of functional connectivity (Figure 2). Gradients also need to be computed from individual connectivity matrix and then aligned post-hoc to each other or to a reference with Procrustes alignment, which poses a challenge if functional connectivity is generated from different parcel definitions. The neuroimaging field has some popular choices of parcellations but there is still no consensus. In addition, age-specific and individual-specific parcellation optimized for the data are becoming more common.
> >
> > Please see also a more extended response to reviewer ghC6.

---

> > > ### Author Response · Authors · 2024-11-26
> > > **Response to reviewer GbuD (part3)**
> > >
> > > # Prediction of behavioral phenotypes
> > > > "I think the paper would substantially improve to consider prediction of external information such as demography and cognitive abilities available in the HCP cohort to ground the methodology’s utility more quantitatively in terms of such ground truth information available for the individuals. It would in this context also be possible to understand and compare the proposed seed-based beta-VAE compressions utility when compared to standard neuroimaging compression methodologies operating directly on the spatio-temporal data for which there is a large literature using various approaches to predict aspects of the individuals of neuroscientific interest."
> > >
> > > We have conducted these analyses and included the relevant results in the Appendix A14: prediction of behavioral phenotype in individuals.
> > >
> > > However, we don't think that this analysis is superior than our individual identification analysis in Appendix A13, nor it would solve your concern "a poor model focusing on noisy signals may have high subject consistency as noise/bias may be subject specific, produce high degree of homogeneity and lend itself well reconstructed as such noise confounders may be prominent as fMRI generally suffers from poor SNRs. As such, the methodology is not compared to ground truth information of neuroscientific interest such as recovery of task responses in task data, ability to predict properties of the individuals such as age, gender and cognitive capabilities etc". Because if you believe that there is noise in the fMRI data that is subject-specific, then the "noise" would also be related to the behavioral phenotype such as age and sex. While non-neuronal subject-specific factors might contribute to the functional connectome reliability such as the head motion and respiratory patterns [9], this issue is present in the data regardless of the model used, and should be tackled with acquisition and data preprocessing strategy. Therefore, this confound should not be penalizing against our framework. More importantly, we demonstrated the separate of functional networks, which is consistent with neuroscientific knowledge.
> > >
> > > [9] Power, J.D., Lynch, C.J., Adeyemo, B., Petersen, S.E., 2020. A Critical, Event-Related Appraisal of Denoising in Resting-State fMRI Studies. Cerebral Cortex 30, 5544–5559. https://doi.org/10.1093/cercor/bhaa139

---

> > > > ### Comment · Reviewer_GbuD · 2024-11-29
> > > > **I appreciate the authors efforts addressing my concerns but I am inclined to maintain my score**
> > > >
> > > > I thank the authors for their careful rebuttal and added experimentation. I highly appreciate the substantial work put into addressing my concerns and including additional analyses by PCA and ICA on the raw time series data as well as including prediction of ground truth information, i.e. gender and age in the supplementary material. While I agree that such ground truth predictions can also be influenced by confounders, my concern remains that the proposed approach using a \beta-VAE does not really demonstrate clear benefits as opposed to conventional modeling procedures using PCA and ICA. Methodologically, the approach is straightforward and not of high novelty and as the results I find rather unclear in terms of merits of the proposed procedure over such simple conventional linear dimensionality procedures I find the approach unconvincing.

---

> > > > > ### Author Response · Authors · 2024-11-30
> > > > > **Asking for clarifiication on conventional procedures**
> > > > >
> > > > > Dear reviewer,
> > > > >
> > > > > Thank you for your feedback. We respect your decision to maintain the score, but I was just wondering if you would kindly point out some citations for "conventional linear dimensionality procedures", this can help us with our resubmission process. I suspect you mean dimensionality in the spatial-temporal domain, but I am unsure. We showed the benefit of direct seed map embedding for its conserved scale between -1 and 1 across data, the intuitive back-projection to the original neural space for seed map representations even for group-averaged or location-averaged summary data. If this refers to the results that "autoencoder-based methods are no better than linear methods for embedding seed maps", autoencoder-based models seem to be more effectively capturing the fine details in the data with lower dimensions (more efficient), which is the main benefit of the model. Moreover, if the problem space is simple enough, the more complex model may not have many additional benefits. It is still an important message to convey since other reviewers seem to care about more complex models used by prior research. We respect your right to have different opinions or to think this kind of message is not what this conference is for, but just want to confirm that our interpretation of your ideas is correct and the kind of "conventional" methods that you are specifically referring too. Thank you again for spending the time to read our paper and help us improve.

---

> > > > > > ### Comment · Reviewer_GbuD · 2024-12-02
> > > > > >
> > > > > > Thanks for getting back for clarification. Yes, you are interpreting my comments correctly and I do mean dimensionality in the spatial-temporal domain. I also agree that AEs generally have the benefit especially when using very low dimensionality that they can compress more efficiently.

---

### Official Review · Reviewer_ghC6 · 2024-11-01

**Soundness:** 3
**Presentation:** 4
**Contribution:** 3
**Rating:** 8
**Confidence:** 4

**Summary:**

In this work, the authors present an approach for learning a compact representation of functional connectivity data which generalize across different parcellations. The authors utilize a convolutional variational auto-encoder (VAE) and assess the quality of embeddings compared compact representations across multiple data sets.

**Strengths:**

The paper is well-written and fairly clear, and the approach seems novel as far as I have found.

Despite some missing pieces such as cross-validation results and error bars, the greatest strength of this work is in the extensive experimentation and results, which include multiple types of validation including reconstruction errors, distribution of canonical networks, and inter and intra-subject varaibility. The results in this study are quite convincing, and the method seems sound given its simplicity. The authors take pains to provide sufficient details for replicability in terms of model architecture and optimization, and a significant amount of work is included in the appendix which elaborates on the results and justifies hyper-parameter discussions.

I want to take the simplicity of this approach as a strength rather than a weakness as well. I find that simple approaches are refreshing and if they outperform state of the art, should be widely adopted. That said, as I elaborate on in the weaknesses section, convolutional VAEs are a fairly old model and comparisons ought to be made with more novel methods such as Transformer-based architectures (such as Brain LM [1] perhaps), or generative adversarial networks (GANs) [2]. That said, the model convincingly outperforms the linear methods used as baselines in this work.

[1] Ortega Caro, Josue, et al. "BrainLM: A foundation model for brain activity recordings." bioRxiv (2023): 2023-09.

[2] Goodfellow, Ian, et al. "Generative adversarial networks." Communications of the ACM 63.11 (2020): 139-144.

**Weaknesses:**

While the simplicity of the model could be a strength, more work is needed to justify why a convolutional VAE would outperform other nonlinear reconstruction methods which exist in the literature. The most obvious omission is that the authors ought to compare against a traditional convolutional auto-encoder in order to justify why the variational model should be used in favor of a non-variational approach. As I've mentioned above, more novel reconstruction methods such as GANs and Transformer-based architecture ought also to be considered. The performance above linear reconstruction methods is convincing; however, multiple nonlinear reconstruction methods exist and have been applied to functional connectivity before [3,4,5]. This omission by itself puts this below the acceptance threshold for me, and unless the authors can provide a substantial rebuttal which includes comparisons, I would encourage them to resubmit this work at a later date given more substantive comparisons to modern architectures.

Furthermore, it seems the authors have not performed any cross-validation or multiple model training (across different random initializations) in order to reduce the variance from individual runs of the model. It is standard practice to perform a k-fold cross validation and provide error bars across folds in order to assure that model improvements do not amount to a particularly good subset of data or model initialization. This omission is more glaring than the previous one and puts it below a marginal reject to a full reject for me. If the authors can provide k-fold cross validation results in their revision, I will consider improving my score to a marginal reject; however, I think this work would benefit from a substantial revision and resubmission at a later date.

Finally, I think this work would benefit from training on a larger cohort of data, such as the UKBiobank [6] which contains several thousand participants. This alone does not lower the score for me; however, I will point out that the data sets in this study are quite small and finding a larger cohort of data would substantially improve the results here.

[3] Zhang, Lu, Li Wang, and Dajiang Zhu. "Recovering brain structural connectivity from functional connectivity via multi-gcn based generative adversarial network." Medical Image Computing and Computer Assisted Intervention–MICCAI 2020: 23rd International Conference, Lima, Peru, October 4–8, 2020, Proceedings, Part VII 23. Springer International Publishing, 2020.

[4] Zhao, Jianlong, et al. "Functional network connectivity (FNC)-based generative adversarial network (GAN) and its applications in classification of mental disorders." Journal of neuroscience methods 341 (2020): 108756.

[5] Zuo, Qiankun, et al. "Brain Functional Network Generation Using Distribution-Regularized Adversarial Graph Autoencoder with Transformer for Dementia Diagnosis." Computer modeling in engineering & sciences: CMES 137.3 (2023): 2129.

[6] Bycroft, Clare, et al. "The UK Biobank resource with deep phenotyping and genomic data." Nature 562.7726 (2018): 203-209.

**Questions:**

1) how does the convolutional VAE compare with other nonlinear (deep-learning based) reconstruction methods? For example GANs or transformer-based architectures?

2) how does the performance of the model vary across folds in the data or across multiple model initializations?

---

> ### Author Response · Authors · 2024-11-25
> **Response to reviewr ghC6**
>
> Dear reviewer,
>
> Thank you for your extensive review of our paper. I would like to respond to some of your major concerns below. I understand that this might not be your favorite paper, but I would greatly appreciate it if you could provide me with feedback on whether I have understood and addressed your concerns and what additional areas for improvement are needed. This is my first submission to ICLR, and I would like your honest opinion on whether a resubmission to the same or a similar venue with some revisions of the original paper is appropriate, or if you suggest other venues that might be a better fit for the content. This is also my first submission of a paper on Open Review, so please excuse me and point out potential improvements for my response formatting.
>
> ## Justification for Variational Autoencoder Model and Comparison to Alternatives:
>
> > "The most obvious omission is that the authors ought to compare against a traditional convolutional auto-encoder in order to justify why the variational model should be used in favor of a non-variational approach."
>
> We would like to re-emphasize our goal to obtain a continuous latent space that is likely to disentangle generative factors and generalize to new data, rather than the most accurate reconstruction.
>
> Conventional autoencoder would not provide the same continuous, regularized latent space to give an intuitive sense of variation across the latent space would look like (Figure 2). They may “fracture the manifold into many different domains” and “result in very different codes for similar images" [1]. We conducted additional analysis to obtain the conventional autoencoder model (by setting $\beta$ = 0 so that only reconstruction error contributes to the total loss) and updated the figures (Figures 2-4) in our revision draft to experimentally validate this point and demonstrated a much worse disentangled latent space with conventional autoencoder.
>
> > 1.	how does the convolutional VAE compare with other nonlinear (deep-learning based) reconstruction methods? For example, GANs or transformer-based architectures?
>
> We additionally trained a Generative Adversarial Network (GAN)-based autoencoder for reconstruction: adversarial autoencoder [1]. In this model, a discriminator network was trained to regularize the autoencoder latent distribution. We conducted additional experiments and produced results with an Adversarial Autoencoder and updated the figures (Figure 2-4) in the revised manuscript. In general, we found the performance of the adversarial autoencoder to be similar to the VAE ($\beta$ = 1)
>
> [1] Makhzani, A., Shlens, J., Jaitly, N., Goodfellow, I., Frey, B., 2016. Adversarial Autoencoders. ICLR. https://doi.org/10.48550/arXiv.1511.05644

---

> > ### Author Response · Authors · 2024-11-25
> > **Response to reviewr ghC6 (part 2)**
> >
> > ## Comparison to Related Work in Literature (novelty concerns)
> > >The performance above linear reconstruction methods is convincing; however, multiple nonlinear reconstruction methods exist and have been applied to functional connectivity before. This omission by itself puts this below the acceptance threshold for me, and unless the authors can provide a substantial rebuttal which includes comparisons, I would encourage them to resubmit this work at a later date given more substantive comparisons to modern architectures.
> >
> > We agree that a discussion with prior work is crucial to demonstrate the contribution for the work. We now added a detailed discussion in the context of previous work in Section 2: Related work in the revised draft.
> >
> > While a lot of human neuroimaging work uses deep learning to capture fMRI activity, structural connectivity and functional connectivity, many of the work (including the ones sited by you) were using abstract nodes in either independent components or predefined regions, and have one single goal, e.g. to make predictions (on disease classification, prediction of masked brain activity etc.). Moreover, many of them consider the connectome matrix from one subject as a single sample, while our data samples are seed maps with thousands of samples in each subject (related to your question about sample size). These were very different from our goal of obtaining a general, compact latent space for comparing data from different parcels (can be predefined regions or independent components above) with **pre-computed mappings**. Most of the existing work also focused on a specific machine learning benchmark within a single dataset instead of obtaining low-dimensional embeddings and test different benchmarks on multiple datasets. Therefore, we find it hard to directly compare our performance to those models because we could not identify one model that is directly comparable, unlike the other computer vision application with a standard task and metric (e.g., classification with MNIST or CIFAR-10 dataset with different variations of autoencoders).
> >
> > The most similar work in spirit is the representing of functional connectivity data into principal gradients to visualize the dominant spatial modes in functional connectivity [7-8] , but the samples in the latent distribution could not be “back-projected” to the original space to provide an intuitive visualization of the effect of varying the gradient from one end to another on the appearance of functional connectivity (Figure 2). Gradients also need to be computed from individual connectivity matrix and then aligned post-hoc to each other or to a reference with Procrustes alignment, which poses a challenge if functional connectivity is generated from different parcel definitions. The neuroimaging field has some popular choices of parcellations but there is still no consensus. In addition, age-specific and individual-specific parcellation optimized for the data are becoming more common. The existence of different parcellation schemes is evident in the citations provided by you.

---

> > > ### Author Response · Authors · 2024-11-25
> > > **Response to reviewr ghC6 (part 3)**
> > >
> > > Here are some specific comparisions to the citations provided:
> > >
> > > - *Our work V.S. (Zhang, et al. 2020) [3]* – Similar to Brain LM, this paper already did the dimensionality reduction to get fMRI activity time series (in terms of 140 regions defined by the Destrieux atlas). The goal was to predict structural connectivity but from functional connectivity rather than finding a latent representation of functional connectivity. The authors claimed to demonstrate high fidelity of structural connectivity prediction and captured subject-specific features. However, the results only showed a few cherry-picked examples.
> > > - *Our work V.S. FNC-GAN (Zhao, Jianlong, et al. 2020) [4]* – Similar to BrainLM, this paper already did the dimensionality reduction to get fMRI activity time series (in terms of 50 independent components) and the “functional network connectivity (FNC) estimated for each subject based on group-ICA was used as the input of the GAN module”. Each input in their paper is the functional connectome (pairwise correlation across all nodes) for each single subject. This is one level higher in abstraction than our approach, and two levels higher than brainLM because each input in our paper is a seed map (one correlation map for each node). Again, they don’t consider the spatial relationship in their abstracted connectivity matrix. They used the deep neural network for supervised learning to classify patients from controls, which is very different from our goal.
> > > - *Our work V.S. (Zuo et al. 2023) [5] similar to above, this paper takes the whole connectome as input and tried to do classification.
> > > - *Our work V.S. BrainLM (Ortega Caro, Josue, et al.) [6]*– they use a BERT-like transformer architecture to reproduce masked fMRI activities (time series from 400+ abstract nodes) whereas we are dealing with the spatial pattern of functional connectivity (images). Potentially due to the differences in the question, the brainLM reconstruction performance was relatively low (R between 0.185 and 0.280 on two datasets), while our reconstruction performance is relatively high ($\eta^2$ ~ 0.7).  This model would not help us get a parcel-independent representation of functional connectivity.
> > >
> > > [2] Goodfellow, Ian, et al. "Generative adversarial networks." Communications of the ACM 63.11 (2020): 139-144.
> > >
> > > [3] Zhang, Lu, Li Wang, and Dajiang Zhu. "Recovering brain structural connectivity from functional connectivity via multi-gcn based generative adversarial network." Medical Image Computing and Computer Assisted Intervention–MICCAI 2020: 23rd International Conference, Lima, Peru, October 4–8, 2020, Proceedings, Part VII 23. Springer International Publishing, 2020.
> > >
> > > [4] Zhao, Jianlong, et al. "Functional network connectivity (FNC)-based generative adversarial network (GAN) and its applications in classification of mental disorders." Journal of neuroscience methods 341 (2020): 108756.
> > >
> > > [5] Zuo, Qiankun, et al. "Brain Functional Network Generation Using Distribution-Regularized Adversarial Graph Autoencoder with Transformer for Dementia Diagnosis." Computer modeling in engineering & sciences: CMES 137.3 (2023): 2129.
> > >
> > > [6] Ortega Caro, Josue, et al. "BrainLM: A foundation model for brain activity recordings." bioRxiv (2023): 2023-09.
> > >
> > > [7] Margulies, D.S., Ghosh, S.S., Goulas, A., Falkiewicz, M., Huntenburg, J.M., Langs, G., Bezgin, G., Eickhoff, S.B., Castellanos, F.X., Petrides, M., Jefferies, E., Smallwood, J., 2016. Situating the default-mode network along a principal gradient of macroscale cortical organization. Proc. Natl. Acad. Sci. U.S.A. 113, 12574–12579. https://doi.org/10.1073/pnas.1608282113
> > >
> > > [8] Vos de Wael, R., Benkarim, O., Paquola, C., Lariviere, S., Royer, J., Tavakol, S., Xu, T., Hong, S.-J., Langs, G., Valk, S., Misic, B., Milham, M., Margulies, D., Smallwood, J., Bernhardt, B.C., 2020. BrainSpace: a toolbox for the analysis of macroscale gradients in neuroimaging and connectomics datasets. Commun Biol 3, 1–10. https://doi.org/10.1038/s42003-020-0794-7

---

> > > > ### Author Response · Authors · 2024-11-25
> > > > **Response to reviewr ghC6 (part 4)**
> > > >
> > > > ## Cross-Validation and Model Training Variability
> > > > >2.	how does the performance of the model vary across folds in the data or across multiple model initializations?
> > > >
> > > > We agree that it is standard practice for the conventional machine learning to have cross-validation and random initiation to explore model training variability. However, we recognized that it is not a very common for deep learning training with much more data and long training time [6,9,10]. The more common practice is to train on a large amount of sample and test on a small set of sample/additional datasets which is what we are doing here. I did have variability in performance across test samples in Figure 3.
> > > >
> > > > [9] Higgins, I., Matthey, L., Pal, A., Burgess, C., Glorot, X., Botvinick, M., Mohamed, S., Lerchner, A., 2017. β-VAE: LEARNING BASIC VISUAL CONCEPTS WITH A CONSTRAINED VARIATIONAL FRAMEWORK. Presented at the ICLR.
> > > > [10] Kingma, D.P., Welling, M., 2022. Auto-Encoding Variational Bayes. https://doi.org/10.48550/arXiv.1312.6114
> > > >
> > > > Since each sample is a seed map, each training epoch consists of 594200 samples (10% samples of 59412 cortical vertices from 100 subjects), much higher than the ~100 training samples in [5]. I do have the 5-fold cross-validation models with VAE ($\beta$ = 20) ready and the loss profiles look similar to the one with all training data. However, I am not sure if I can produce all the visualizations by the revision deadline. If you are still concerned about "error bars" I would appreciate if you point out which specific analyses would benefit from having the results generated from those different model trainings.
> > > >
> > > > ## Data size and diversity
> > > > > Finally, I think this work would benefit from training on a larger cohort of data, such as the UKBiobank which contains several thousand participants. This alone does not lower the score for me; however, I will point out that the data sets in this study are quite small and finding a larger cohort of data would substantially improve the results here.
> > > >
> > > > It remains possible that different performance metrics could be slightly improved with a large cohort such as the UKBiobank as the training data. However, given that the reconstruction performance is approaching the noise ceiling in capturing individual-specific features in Figure 3, and the functional connectivity spatial pattern is known to be very robust and stereotypical [11], the inclusion of a massive dataset may have diminishing returns and will be challenging especially if you want to have cross-validation and multiple initialization.
> > > >
> > > > [11] Gratton, C., Laumann, T.O., Nielsen, A.N., Greene, D.J., Gordon, E.M., Gilmore, A.W., Nelson, S.M., Coalson, R.S., Snyder, A.Z., Schlaggar, B.L., Dosenbach, N.U.F., Petersen, S.E., 2018. Functional Brain Networks Are Dominated by Stable Group and Individual Factors, Not Cognitive or Daily Variation. Neuron 98, 439-452.e5. https://doi.org/10.1016/j.neuron.2018.03.035

---

> > > > > ### Comment · Reviewer_ghC6 · 2024-11-25
> > > > > **Cross Validation and Model Training Variability**
> > > > >
> > > > > I can accept the reasoning for not performing cross validation due to time limitations; however, this ought to be mentioned explicitly in the discussion of limitations. If you can provide some evidence of performing these experiments even in a table, I will raise my score.

---

> > > > > > ### Author Response · Authors · 2024-11-26
> > > > > > **updated figures**
> > > > > >
> > > > > > Dear reviewer,
> > > > > >
> > > > > > I uploaded my revision draft, please find the relevant sections in section A.7 Figure 9.

---

> > > > > > > ### Comment · Reviewer_ghC6 · 2024-11-26
> > > > > > > **Updating Score based on Revision**
> > > > > > >
> > > > > > > Thank you for posting the revision! I am changing my score accordingly. I would like to recommend this paper for acceptance.

---

> > > > > > > > ### Author Response · Authors · 2024-11-27
> > > > > > > > **Thank you!**
> > > > > > > >
> > > > > > > > Thanks for following up. Have a good day!

---

> > ### Comment · Reviewer_ghC6 · 2024-11-25
> > **Possible to Provide Figures for Revision**
> >
> > These seem like more than reasonable responses to my concerns regarding the lack of comparisons with other nonlinear construction techniques; however, can you provide a table or figure demonstrating the purported revisions so I can verify they were completed? I am more than happy to revise my score for this substantial addition.

---

> > > ### Author Response · Authors · 2024-11-25
> > > **provide figures for revision**
> > >
> > > Dear reviewer, thank you for your reply. Please wait a little bit before I upload my revision pdf. I was originally under the impression that discussions were due before the revision pdf so I was trying to go through the comments first. The pdf will be uploaded by Nov 26.

---

### Meta-Review · Area_Chair_Dpp5 · 2024-12-19

**Metareview:**

This submission contributes a convolutional variation auto-encoder to extract a representation of functional connectivity data. The submission generated interests from the reviewers and discussion. The reviewers appreciated the thorough discussion period. However, it is not clear that the submission meets the high bar of ICLR. The reviewers raised several important points that underscore that the innovation is more in the application than in the machine-learning method, but the application would benefit from more thorough validation. Among other topics, the reviewers suggested a more thorough quantification of the variability as a function of train and test data, and experiments on larger and different data.
The review also revealed a lack of convincing evidence of the benefits compared to classic methods used in neuroimaging. Indeed, the problem of comparing across approaches is best posed for a well-defined task. Metrics used (reconstruction, homogeneity...) are not tasks with a clear neuroscience interest. The results of the prediction of behavioral phenotypes are interesting in this respect, but the results are currently not convincing.

**Additional Comments On Reviewer Discussion:**

There was a good discussion with much back and forth between authors and reviewers. The discussion led to improving the manuscript.

---

### Decision · Program_Chairs · 2025-01-22

Reject